# Multi-omics analysis reveals the dynamic interplay between Vero host chromatin structure and function during vaccinia virus infection
Vrinda Venu [1], Cullen Roth [2], Samantha H. Adikari[3], Eric M. Small [1], Shawn R. Starkenburg [2], Karissa Y. Sanbonmatsu [4,5] & Christina R. Steadman [1] ✉

The genome folds into complex configurations and structures thought to profoundly impact its function. The intricacies of this dynamic structure-function relationship are not well understood particularly in the context of viral infection. To unravel this interplay, here we provide a comprehensive investigation of simultaneous host chromatin structural (via Hi-C and ATAC-seq) and functional changes (via RNA-seq) in response to vaccinia virus infection. Over time, infection significantly impacts global and local chromatin structure by increasing long-range intra-chromosomal interactions and B compartmentalization and by decreasing chromatin accessibility and inter-chromosomal interactions. Local accessibility changes are independent of broad-scale chromatin compartment exchange (~12% of the genome), underscoring potential independent mechanisms for global and local chromatin reorganization. While infection structurally condenses the host genome, there is nearly equal bidirectional differential gene expression. Despite global weakening of intra-TAD interactions, functional changes including downregulated immunity genes are associated with alterations in local accessibility and loop domain restructuring. Therefore, chromatin accessibility and local structure profiling provide impactful predictions for host responses and may improve development of efficacious anti-viral counter measures including the optimization of vaccine design.

While genomic loci can be linearly very distant from one another, they fold into complex configurations in three-dimensional space to enable proximal interaction[1]. These configurations have hierarchical architecture, and along with chemical chromatin modifications, they influence—or are often influenced by—genome functions such as replication, recombination, and gene expression[2]. The flexible, self-organizing, fluctuating nature of the genome is thought to have a major role in its functional plasticity, and several studies have used polymer physics models of chromosomes to predict structure and functionality[3–7]. Recent work demonstrates that a complex, combinatorial, dynamic structure-function relationship exists between chromatin architecture and accessibility, the epigenetic landscape, and genome functionality[8–10]. Deeper empirical evidence from responsive cellular states is required to understand the causality and influence of global versus local chromatin architecture on function. Intrinsic and extrinsic cellular cues alter chromatin architecture, including shifts in proximity during the cell cycle, differentiation[11,12], and in response to infection, drug treatments, and environmental stressors[13–15]. Yet, detailed observations of concurrent alterations in global and local chromatin structure and gene function are still limited. To this end, we investigated the extent of changes in host chromatin interactions, accessibility, gene expression and their correlations using a viral infection model.

Viruses hijack host cellular function, including mechanisms involved in chromatin modification, to enhance viral replication, survival, and proliferation[16]. Studies have demonstrated that many double-stranded DNA viruses (and some RNA viruses) that translocate into the nucleus directly access and interact with transcriptional machinery for replication and host chromosomes for interference of defense programs[17,18]. Non-nuclear viruses, including RNA viruses such as SARS-CoV-2, may impart

¹Climate, Ecology & Environment Group, Los Alamos National Laboratory, Los Alamos, NM, USA. ²Genomics & Bioanalytics Group, Los Alamos National Laboratory, Los Alamos, NM, USA. ³Biochemistry & Biotechnology Group, Los Alamos National Laboratory, Los Alamos, NM, USA. ⁴Theoretical Biology and Biophysics Group, Los Alamos National Laboratory, Los Alamos, NM, USA. ⁵New Mexico Consortium, Los Alamos, NM, USA. ✉e-mail: crsteadman@lanl.gov

significant changes to the host chromatin, including altering gDNA methylation and histone modifications, and in some cases, chromatin structure[8,13,19]. These studies suggest that viral-induced changes to chromatin structure and alteration of host (and virus) epigenetic modifications aids in the progression of infection. However, detailed characterization of host chromatin structure-function dynamics due to non-nuclear viral infection has only been investigated in a handful of cases[8,13]. Given the diversity of infection mechanisms and immunogenicity among various pathogenic viruses, attenuated viral strains, and virus-like particles, a clear understanding of host chromatin dynamics and functional responses would greatly enhance our ability to create counter measures to infections and effective vaccine development. To meet this need, we established a virus infection model using an attenuated vaccinia virus strain to characterize host chromatin structure-function dynamics over the course of viral infection.

The virus family of Poxviridae is comprised of particularly resilient and pathogenic viruses that cause severe infectious disease, including smallpox. Effective eradication of smallpox occurred over forty years ago with an aggressive global vaccination program using vaccinia virus (VACV), a poxvirus with poorly understood origins but efficacious outcomes in humans[20]. The Poxviridae family is unique among dsDNA viruses: they complete their lifecycles entirely in the cytoplasm of host cells[21]. While their large genomic DNA replicates within the host cytoplasm, viral particle release requires protein localization in the host nucleus suggesting interaction with the host genome is required for virulence[22,23]. To establish an experimentally amenable viral infection model to permit interrogation of genomic structure-function relationships, we chose to infect a permissible mammalian cell line (Vero) with an attenuated modified vaccinia virus Ankara (MVA) strain. Vero cell lines, originally derived from African green monkey (*Chlorocebus sabaeus*) kidney cells, are used in many vaccine production platforms[24]. To date, subsets of these cell lines have been considered permeable in varying degrees to VACV and its attenuated strains, including MVA[25,26]. While MVA is non-replicative in most mammalian cells, it expresses all classes of its genes upon cellular entry, synthesizes viral DNA, and produces immature virions that are not released[27]. MVA vectors are promising candidates for their ability to express transgenes and immunogenicity in vivo[25]. Several recombinant vaccines that employ MVA vectors have been developed against diseases including HIV[28], Zika virus[29], and SARS-CoV-2[30,31].

To obtain detailed, high resolution chromatin architecture data, we employed Hi-C, the most recent iteration of chromosome conformation capture sequencing technology, which can illuminate all contacts within a genome at up to 1 kb resolution[32]. Hi-C uses in situ, proximity-based ligation followed by sequencing for genome-wide characterization of 3D chromatin configurations. These include chromatin domains with preferential interactions and distinctive boundaries between actively transcribed euchromatin versus inactive transcriptionally repressed heterochromatin, or A and B compartments, respectively[33]. While simplistic, partitioning of the genome into A and B compartments occurs on a large (MB) scale, and genomic architectural rearrangements occur in at least 36% of the genome during differentiation, resulting in distinctive juxtaposition pattern of A and B compartments[11]. Recent studies have demonstrated that chromatin accessibility assays complement Hi-C contact maps, and *cis* proximal loci may have similar accessibility, suggesting that higher-order structure may influence function[34]. Mapping of chromatin accessibility may be performed using Assay for Transposase Accessible Chromatin by Sequencing (ATAC-seq) which provides information on nucleosome-free regions. The ease of this method underscores its relative ubiquity in the epigenomics field.

Here, we attempt to overcome the seemingly mutually exclusive nature of genome architecture variation and fine-tuned resolution analysis at the gene level by pairing Hi-C, ATAC-seq, and transcriptomics. Using these analyses, we investigated the extent of host chromatin structure and function changes in response to a viral agent without direct contact in the nucleus. We found that MVA infection alters global chromatin organization, including increased chromatin contact frequency and reduced chromatin accessibility as infection progressed over a 24-h period. Though not linear, we detected close association of fine-scale chromatin accessibility with gene expression and chromatin loop formation. Changes in gene expression that involved downregulation of certain proinflammatory genes were concurrent with restructuring of chromatin loop domains and local accessibility reduction. These findings underscore the dynamic plasticity and correlation of both genome structure and function providing evidence that structure modulates but is not deterministic of function in response to viral infection.

## Results

To investigate concurrent alterations in host chromatin accessibility and 3D organization as viral infection progresses, we infected a susceptible cell line (Vero), derived from African green monkey, with MVA (Fig. 1a). Using immunofluorescence microscopy, we confirmed successful infection and its gradual progression at 12-, 18-, and 24-hours post-infection (hpi) (Fig. 1b, Fig. S1). At each time point, cells were collected from mock-infected (control) and MVA-infected cell culture flasks for paired measurement of 3D genomic organization, chromatin accessibility, and gene expression using Hi-C, ATAC, and RNA sequencing, respectively (Fig. 1a). The use of time-matched cultures control for inherent changes in chromatin structure or function related to time spent in culture that are unrelated to infection progression; further, cell viability was recorded as infection progressed. We generated and sequenced 12 Hi-C libraries, with each replicate per condition totaling over 600 million 150-bp read pairs. We detected approximately 700 million valid Hi-C contacts per condition per time point (12, 18, and 24 hpi) (Table S1). Twelve ATAC-seq libraries and 12 RNA-seq libraries (both paired to the aforementioned Hi-C libraries) were generated and sequenced to around 40 million (Table S2) and 17 million read pairs per library, respectively (Table S3). We observed high replicate concordance in all three datasets (Fig. S2); thus, replicates were combined for visualization. For all other analyses, replicates were kept separate unless specified otherwise. Principal component analysis of ATAC-seq read pileup, Hi-C contact frequency, and gene expression (Fig. 1c, d, e, respectively) demonstrated distinct separation of mock-infected control cultures from MVA-infected cultures, indicating global changes in host chromatin architecture, accessibility, and expression due to viral infection. Using these high-resolution paired Hi-C, ATAC- and RNA-seq datasets with matching mock-infected controls at each time point, we report concurrent changes in chromatin architecture and accessibility as viral infection progressed in this model system and functional outcomes of MVA infection.

### MVA infection reorganizes host chromatin architecture

Analysis of Hi-C sequencing data revealed substantial alteration of chromatin contacts in response to MVA infection. For example, chromosome 27 displayed some of the most pronounced changes in chromatin architecture as visualized via Hi-C contact maps (Fig. 2a) and contact ratio maps (Fig. 2b). Contact maps compare mock-infected controls (lower diagonal) to MVA-infected cells (upper diagonal) at each time point (Fig. 2a). Fold change profiles of contacts (Fig. 2b) display increased intensity (red color) in the off-diagonal region and the lack of color on the diagonal proximal region indicating more alterations in long-range intra-chromosomal contacts compared to short-range contacts.

The healthy Vero genome is clearly organized into areas of chromatin activity denoted as A (active) or B (inactive) compartments. As viral infection progressed, the number of genomic regions exchanging compartments increased indicating global changes in genome organization. Compartment assignments were calculated using a Pearson correlation coefficient for mock-infected control and MVA-infected Hi-C maps (Fig. 2c). Representative large-scale compartment exchanges are visualized at 100 kb resolution, where highlighted areas (red boxes)

indicate regions of exchange as infection progressed (Fig. 2c). The strength of this compartment exchange, indicative of complete versus weak exchange, increased as infection progressed. To quantify compartment exchange, we identified co-variate regions in Hi-C data at 100 kb resolution using principal component analysis. Using ATAC-seq data, we assigned co-variate regions as activate (A) or inactive (B) compartments based on high or low accessibility, respectively. We found the number of regions strongly exchanging compartments (A to B, B to A) considerably increased as infection progressed (Fig. 2d). For all contacts assessed, 2.4% of the Vero genome exchanged compartments at 12 hpi, 7.8% at 18 hpi, and 12% at 24 hpi, with a slight bias toward A to B exchange (Fig. 2d). Compartment exchange occurs in response to a number of biological processes, including viral infection; here, these findings suggest that MVA infection moderately alters the compartment profile of the Vero genome into a less 'active' mode with greater propensity toward B compartmentalization.

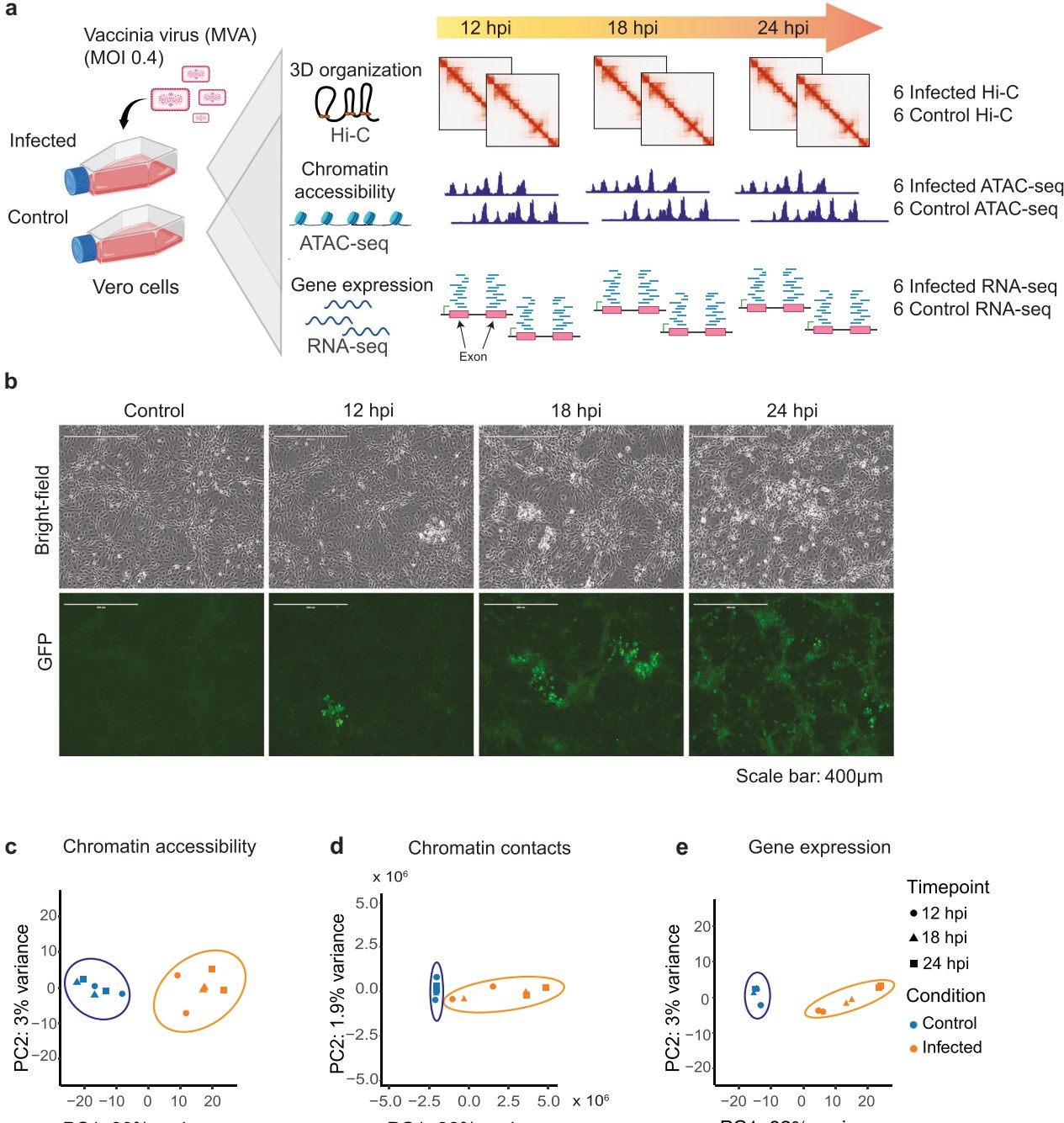

**Fig. 1 | Infection model system reveals changes in chromatin architecture, accessibility, and gene expression over time. a** Schematic representation of the experimental design where Modified Vaccinia virus Ankara (MVA) was used to infect Vero cells with an MOI of 0.40. Figure cartoon created with Biorender. Vero cells cultivated in the same flask were divided for Hi-C, ATAC-seq, and RNA-seq experiments to provide paired data sets for direct comparison. Cells from mock-infected (control) and MVA-infected cultures were collected at 12, 18, and 24 hours post infection (hpi) with two biological replicates each. A total of 12 Hi-C libraries, 12 ATAC-seq libraries, and 12 RNA-seq libraries were prepared. **b** Infection efficiency was verified by immunofluorescence using an anti-vaccinia virus monoclonal antibody (green). Cell density and morphology can be seen in the bright-field images. **c** Principal component analysis of genome-wide chromatin accessibility from ATAC-seq data, **d** Chromatin contacts from Hi-C sequencing data, **e** Gene expression levels from RNA-seq data demonstrate clear separation between mock-infected control and MVA-infected cultures.

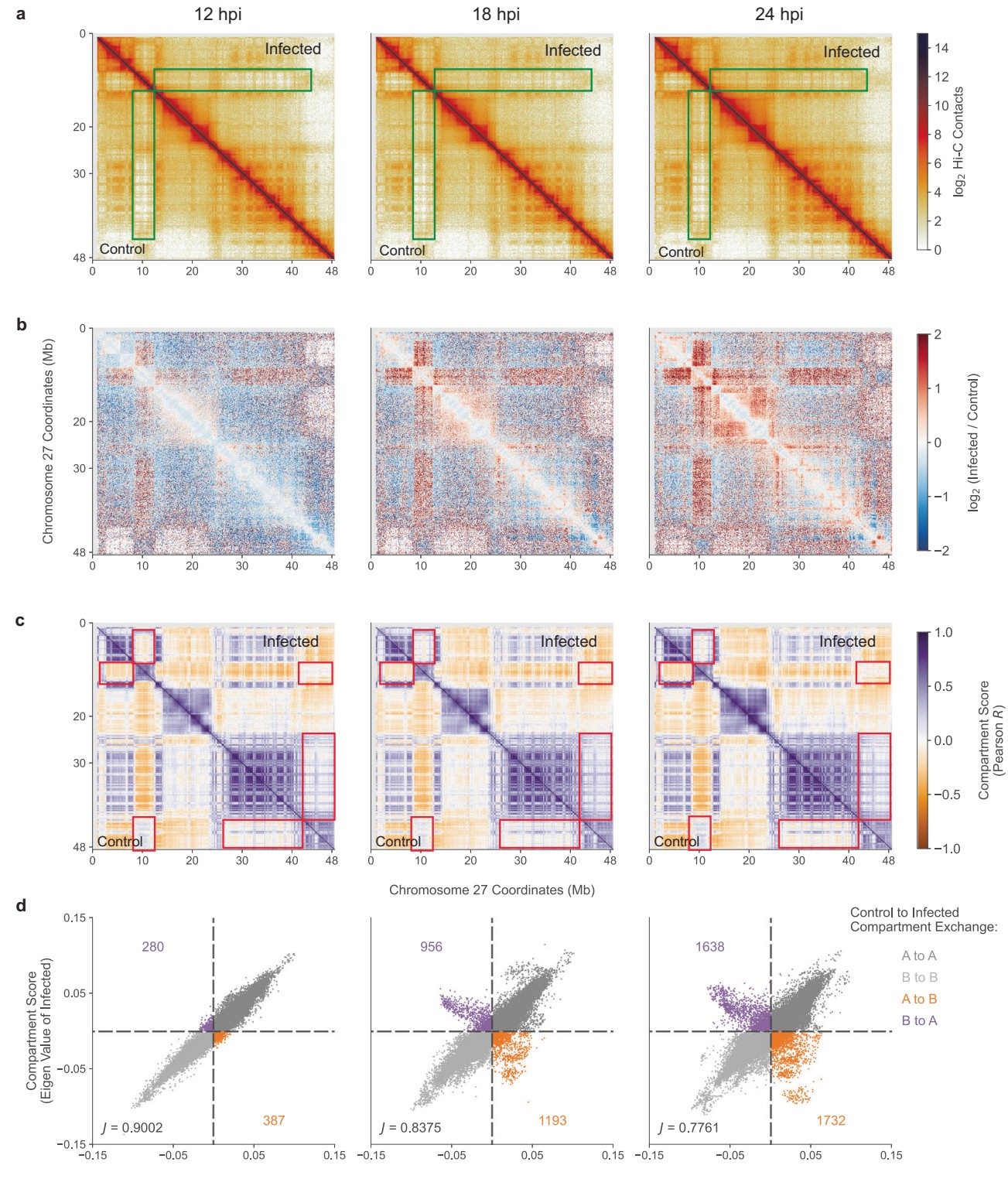

**Fig. 2 | Modified Vaccinia virus Ankara (MVA) infection induced progressive changes in Vero host chromatin 3D architecture. a** Hi-C contact matrices of chromosome 27 in mock-infected control (lower diagonal region) versus MVA-infected (upper diagonal region) cells at 12, 18, and 24 hours post infection (hpi). Green boxes indicate regions with pronounced contact differences between control and MVA-infected cells, where the bin size is 100 kb. **b** Differential (infected/control) contact matrices of chromosome 27 across all three time points are shown at 100 kb resolution. Blue indicates control-biased contacts, where more contacts were observed in control cells, and red indicates infection-biased contacts, where more contacts were observed in MVA-infected cells. **c)** Pearson correlation matrix of chromosome 27 in mock-infected control (lower diagonal region) versus MVA-infected (upper diagonal region) cells, where bin size is 100 kb. Red boxes highlight the regions with altered compartmentalization. **d** Scatterplots demonstrating replicate averaged compartment scores (eigen value) of control vs infected cells in 100 kb bins across the genome at all three time points. Off-diagonal points represents the compartment exchange events, where orange indicates A to B compartment exchange and purple indicates B to A compartment exchange. Counts of compartment exchange are annotated along the off diagonal. The Jaccard index is annotated in the bottom left and measures the overlap between compartment assignments between mock-infected control and MVA-infected scores.

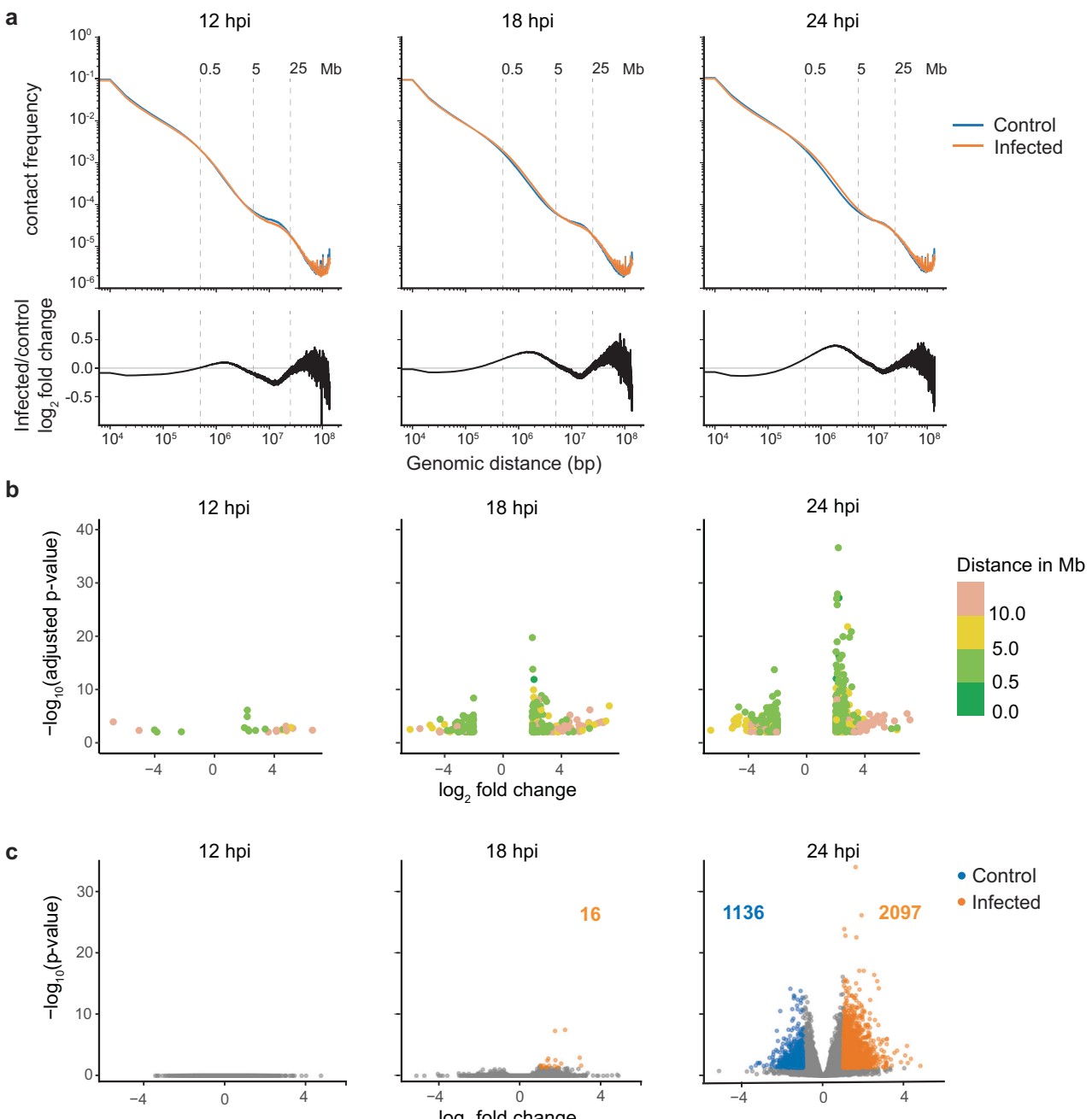

**Fig. 3 | MVA infection predominantly alters mid- to long-range chromatin interactions. a** Contact frequency in control (blue) and MVA-infected (orange) cells as a function of genomic distance in log scale is plotted for all three time points; replicates are combined. Bottom panel: log2 fold change in contact frequency between control and infected cells is plotted as a function of genomic distance in log scale. **b** Genomic regions with significant differences in the number of contacts between mock-infected control and MVA-infected cultures were identified using multiHiCcompare at 100 kb resolution and plotted as log₂ fold change in contact frequency (x-axis) and negative log₁₀ adjusted p-value (y-axis). Color scale annotates the distance between contacting regions. Points in the positive axis represent infection-biased regions (more contacts due to MVA-infection than in control) and points in the negative axis represent control-biased regions. **c** Volcano plots represent identified differential loops between control and MVA-infected cells at all three time points. Gray dots represent all called differential loops where significant (adjusted p-value < 0.05) control-biased loops (log₂ fold change < -1) are shown in blue and infection-biased loops (log₂ fold change > 1) are shown in orange.

## Infection increased mid- to long-range intra-chromosomal interactions

To further understand the changes in 3D genome organization, we assessed the frequency of contacts among chromatin from Hi-C data, which provides a direct measure of spatial interaction between any two points in the genome at a given time[35]. From this analysis, we found that mid- to long-range chromosome interactions increased as MVA infection progressed across 12, 18, and 24 hpi time points (Fig. 3a). This was determined from plotting the contact frequency as a function of the distance between regions in contact, where we observed the greatest difference in the number of the contacts in the 500 kb to 5 Mb range with a bias toward increased contacts in MVA-infected cells. This difference is clearly visualized from the contact frequency fold change of the genomic distance (Fig. 3a; bottom panel). We corroborated this finding using the multiHiCcompare[36] tool, which normalizes contact frequency in 100 kb bins across each chromosome and detects infection- and control-biased contact regions (Fig. 3b). We denote

'infection-biased' bins as regions where MVA infection increased chromatin interactions more frequently than interactions observed in mock-infected control cells; similarly, 'control-biased' bins represent regions where MVA infection induced a loss of chromatin interactions compared to control cells. From this broad scale analysis, there were more interactions in the mid- to long-range distance (500 kb to 5 Mb, light green colored points in Figure 3b) in MVA-infected cells compared to controls. We further observed a significant increase in long distance contacts as infection progressed. At 24 hpi, approximately 66% (479) of the identified significantly different bins (726) represented mid- to long-range distance interactions. However, in a fine-scale analysis (in 10 kb bins across each chromosome), less than 28% (68) of the significantly different bins represent mid- to long-range distance interactions, whereas 71% of the bins (169) represented short-range distance (<500 kb) interactions (Figure S3a). Relatively fewer bins with significant contact differences were observed across the time course in both mock-infected control and MVA-infected cells (Figure S3 b, c. Only broad-scale analysis results are shown). These analyses demonstrate that intra-chromosomal contacts were increased after viral infection across the entire Vero genome. Furthermore, the pattern of inter-chromosomal contacts was dependent upon individual chromosome behavior, where increased interactions were observed with specific combinations of chromosomes. These included interactions among chromosomes 4, 10, 19, 21, and 26 with 31; 9 and 19; 22 and 26 (Fig. S4). However, there was a decrease in contacts among most chromosome pairs (Figure S4) at 24 hpi. This observation is contrary to that reported for SARS-CoV-2 infection of A549 cells[13], suggesting that MVA infection increased compactness of each chromosome potentially resulting in shrinkage of chromosomal territory and thereby weakening inter-chromosomal interactions. We speculate that the observed predominantly mid- to long-range intra-chromosomal interaction changes likely represent sub-compartment repositioning, whereas the relatively low abundant short range interaction changes represent loop reorganization.

Chromatin loops co-localize regulatory elements with their genomic targets to bring one-dimensionally distal chromatin regions in proximity to one another via loop extrusion mechanisms[37]. These loop structures can be identified from Hi-C contact maps as concentrated contact points marking loop anchors, where the two segments of chromatin are co-localized. We identified approximately 14,000 loops per sample by combining loops called at five, ten, and 25 kb resolution (Table S4). Viral infection did not have a significant impact on loop structure compared to controls at early time points. This was determined from differential loop analysis comparing the contact frequency within loop anchor points between MVA-infected and mock-infected control cells at each time point. 'Infection-biased' loops were defined as those with more contacts in MVA-infected cells than in mock-infected controls: indeed, zero, 16, and 2097 infection-biased loops were identified at 12, 18, and 24 hpi, respectively. However, we did observe some loss of loops due to MVA infection (1136 control-biased loops at 24 hpi) (Fig. 3c). This may indicate moderate fine-scale chromatin remodeling with prolonged infection. However, of the chromatin loops identified across the time course, infection-biased differential loops represent less than three percent, while control-biased loops constitute less than two percent of all the loops called. No significant differences in loops were detected among mock-infected control cells across time points or among MVA-infected cells across time points (Fig. S5).

**Infection reduced chromatin accessibility genome wide**
After determining the extent of host chromatin 3D reorganization in response to MVA infection, we next investigated the fine-scale changes in chromatin accessibility across the genome using the paired ATAC-seq datasets generated from the same cell cultures used for the Hi-C experiments. A total of 300,667 distinct open chromatin regions (OCRs) spanning approximately 5.1% of the host genome were identified. MVA infection notably reduced global accessibility compared to mock-infected controls at 12, 18, and 24 hpi as demonstrated in heat maps (Fig. 4a). Further, as infection progressed from 12 to 24 h, global accessibility considerably

decreased in MVA-infected cells. Differential OCRs increased as infection progressed as well, determined by comparing read coverage at each individual OCR between mock-infected control and MVA-infected cells (Fig. 4b). At each time point, differential OCRs were dominated by regions with reduced accessibility in MVA-infected cells, particularly from 12 to 18 hpi (Fig. 4b) concurrent with reduced global accessibility. Further, significant accessibility changes were also detected from 12 to 24 hpi in MVA-infected cells and were dominated by OCRs with a reduction in their accessibility (Fig. S6). This analysis suggests that the OCRs found in both conditions did not significantly change across the time points; rather, the accumulation or loss of OCRs contributed to the differential accessibility we observed. Collectively, global host chromatin accessibility was reduced as MVA infection progressed in Vero cells.

To determine if the fine-scale accessibility changes were correlated with large-scale compartment exchange, we determined the overlap between differential OCRs with our previously identified A and B compartments. The majority of the differential OCRs (~80.3%) in MVA-infected cells overlapped with the compartments that remained active (A), and a substantial number (~8.4%) overlapped with compartments that remained inactive (B) (Fig. 4c). While the greatest number of differential OCRs were identified at 24 hpi, approximately 10.5% of control-biased OCRs and 7% of infection-biased OCRs overlapped with exchanging compartments (Fig. 4c). Regions with increased accessibility in control cultures are denoted as 'control-biased' OCRs and regions with increased accessibility in MVA-infected cultures as 'infection-biased' OCRs. Of the small fraction of differential OCRs that overlapped exchanging compartments, the majority of control-biased OCRs in MVA-infected cells were within compartments that exchanged from active (A) to inactive (B) compartments. This suggests that these open regions became less accessible and simultaneously moved to B compartments. Conversely, infection-biased OCRs in MVA-infected cells were within compartments that exchanged from inactive (B) to active (A) compartments (Fig. 4c). Overall, the global accessibility change due to MVA infection was primarily independent of large-scale compartment exchange, except for less than 10% of the differential OCRs.

Next, we investigated whether chromatin accessibility change was associated with fine-scale chromatin organizational features such as chromatin loops by examining the ATAC-seq read coverage across differential loop anchor points (Fig. 4d). We found correlations between loop formation and accessibility in both MVA-infected and mock-infected control cells: MVA-infected cells had reduced accessibility at the anchor points of weakened loops (i.e., control-biased loops) (Fig. 4d, left panel). Conversely, accessibility was retained to the same extent as controls at anchor regions of loops that were gained after MVA infection (Fig. 4d, right panel). Thus, chromatin accessibility is potentially a pre-requisite for loop anchoring; however, loop re-organization does not seem to alter accessibility of an otherwise open region. This assertion is based on evidence that control cells retained accessibility at infection-biased loop anchor points, which had more contacts in MVA-infected cells than in controls (as depicted in schematic in the bottom panel of Fig. 4d).

Given that a relationship exists between accessibility and expression[38–40], we sought to predict gene regulatory pathways that were potentially impacted by MVA infection. We focused on strong OCRs (annotated specifically to gene promoters) and categorized the genes as either 'activated' or 'suppressed' by MVA infection based on the overlap of the promoter with an infection-biased differential peak or a control-biased differential peak, respectively. We used KOBAS[41] gene enrichment analysis and found enrichment of several signaling pathways in the 'activated' category in mock-infected control cells, suggesting potentially 'suppressed' outcomes in MVA-infected cells. These pathways included pro-inflammatory NF-κB signaling pathway, TLR signaling pathway, and cytokine-cytokine interaction pathway, among others. Conversely, we identified pathways in the 'activated' category in MVA-infected cells suggesting potential upregulation that included various cell-to-cell signaling pathways such as Rap1 signaling pathway, Ras signaling pathway, and MAPK signaling pathway (Figs. S7, S8).

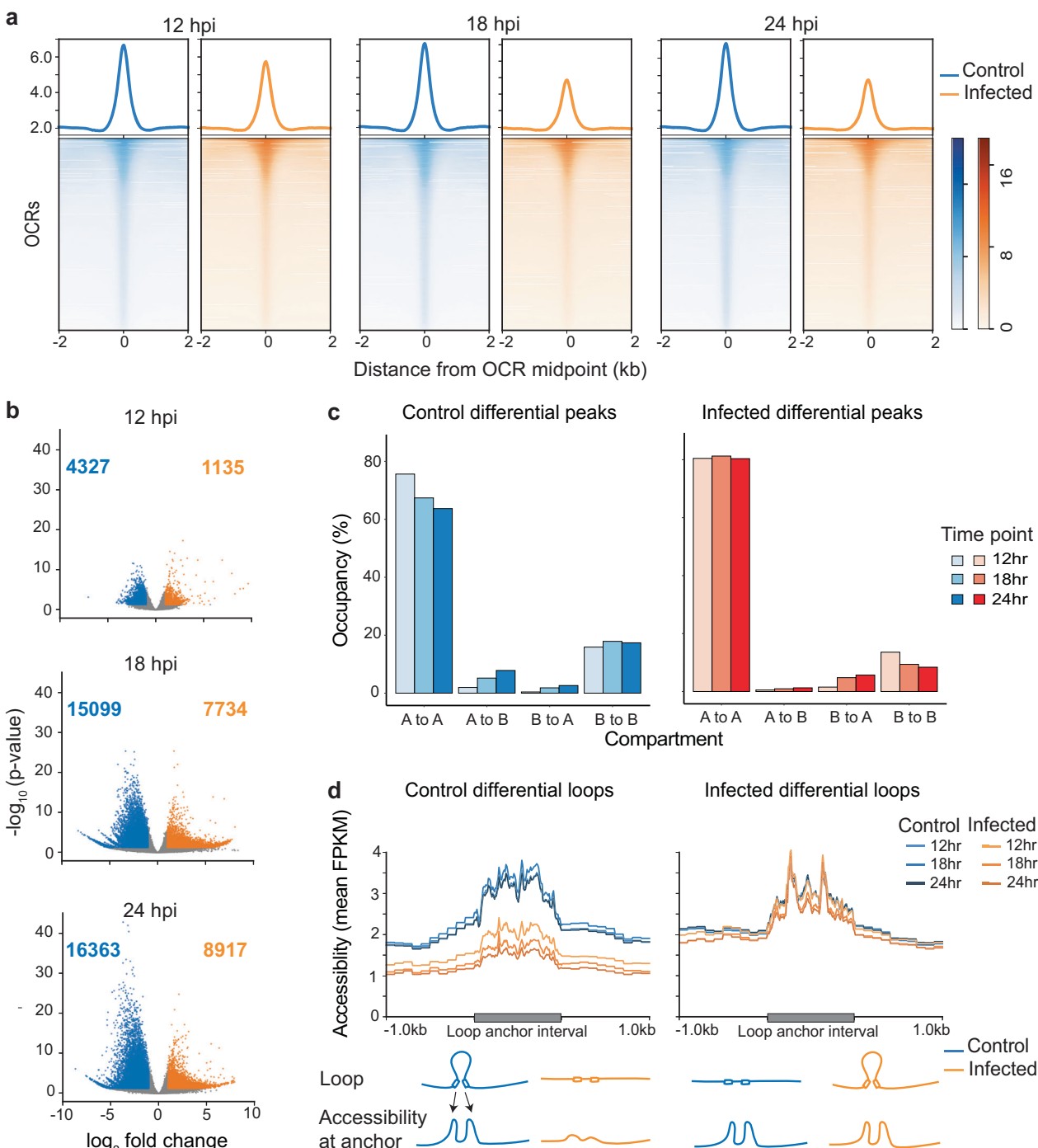

**Fig. 4 | A global decrease in chromatin accessibility in response to viral infection.** **a** Chromatin accessibility across consensus open chromatin regions (OCRs) is plotted for each time point 12, 18, and 24 hours post infection (hpi), where blue indicates mock-infected control cells and orange indicates MVA-infected cells. The ATAC-seq reads from all biological replicates are combined. The normalized average read count profile is plotted in the top panel, while the bottom panel shows the normalized read count per OCR, where each row represents one consensus OCR. Read counts were determined at ± 2 kb region around all consensus OCRs. **b** Volcano plots demonstrate differentially accessible OCRs between mock-infected control and MVA-infected cells at each time point (12, 18 and 24 hpi). Gray dots represent all OCRs. Blue dots indicate significantly different (adjusted *p* value < 0.05)

control-biased OCRs (log$_2$ fold change < −1), while orange dots indicate significantly different infection-biased OCRs (log$_2$ fold change > 1). Control-biased OCRs are regions that are accessible in control cells, while infection-biased OCRs are regions that are accessible in infected cells. **c** Bar plots depict the percentage overlap between mock-infected control (left) and MVA-infected (right) biased differential OCRs and various compartment categories at all three time points. **d** Time- and condition-wise chromatin accessibility (replicate combined normalized ATAC-seq read pileup) across control-biased (left) and infection-biased (right) loop anchor intervals ± 1 kb is plotted in the top panel. A schematic describing the observed relationship between loop anchor points and accessibility is shown in the bottom panel.

## MVA infection altered gene expression correlated with chromatin accessibility

To obtain deeper insight into the functional relationship between chromatin architecture and accessibility, we quantified functional changes that occurred in response to MVA infection using paired RNA-seq datasets generated from the same cell cultures used for Hi-C and ATAC-seq experiments. As expected, we observed more genes with altered expression in MVA-infected cultures compared to mock-infected control cells as infection progressed (Fig. 5a). A greater number of genes were upregulated in MVA-infected cells than in mock-infected controls at all three timepoints (12, 18, 24 hpi), with 836 genes with increased expression at 24 hpi. We used a standard threshold of fold change and $p$ value ($\log_2$ fold change > 2, $p_{adj} < 0.05$) for this analysis, but when we considered all genes with significant differential expression, irrespective of their fold change, we found 4146 downregulated genes and 3959 upregulated genes out of 32,054 total genes at 24 hpi. Thus, there was a marginal increase in the number of downregulated genes at all three time points, suggesting that MVA infection induced changes in host gene expression in both directions with the greatest differential expression, ~25% of genes, at 24 hpi.

Though chromatin accessibility change is not always concordant with changes in transcription[42], they are often associated with one another, either as a permissive factor or an effect of permissiveness[40]. Therefore, in the ATAC-seq and RNA-seq paired datasets, we investigated the chromatin accessibility around the transcription start sites (TSS) of significantly ($p_{adj} < 0.05$) differentially regulated genes at 24 hpi, where the fold change was greater than two. MVA-infected cells had considerably lower accessibility at the promoters of genes that were downregulated compared to mock-infected control cells (Fig. 5b). Conversely, changes in chromatin accessibility around the TSS of upregulated genes were marginally different in MVA-infected cells compared to control cells. This suggests that either increased accessibility is not a prerequisite for increased gene expression or increased gene expression does not elicit a significant change in accessibility. However, decreased gene expression is strongly correlated with decreased accessibility in response to MVA infection.

To determine if alterations in chromatin accessibility are a proxy for functional changes, we investigated whether genes with differential OCRs in their promoters had significantly different gene expression (Fig. 5c). Only OCRs from merged bins of differential ATAC-seq peaks with a base mean greater than 100 and a fold change greater than one localized ± 500 bp around TSS were used. At 24 hpi, the majority of the genes with infection-biased accessibility (increased accessibility compared to controls) had increased gene expression in MVA-infected cells (orange filled circles in Fig. 5c). Control-biased accessibility (reduced accessibility compared to mock-infected controls) had decreased gene expression in MVA-infected cells (filled blue circles in Fig. 5c). Both findings suggest a distinct relationship between accessibility and expression; however, we also detected upregulated genes with decreased accessibility (blue open circles in Fig. 5c) and downregulated genes with increased accessibility (orange open circles in Fig. 5c). Thus, the relationship between accessibility and gene expression, while strong, is not demonstratively linear. This finding corroborates other recent studies reporting that changes in accessibility cannot directly predict changes in gene expression; however, different categories of genes, including housekeeping genes, may have different relationships with accessibility and underscore the variability in this relationship.

Gene expression ultimately dictates how a host cell responds to viral infection. After we identified differential gene expression (both up and downregulation), we investigated the regulatory pathways that may be activated or suppressed in response to infection. We focused on significant differential genes with an adjusted $p$-value less than 0.05 and a $\log_2$ fold change greater than two and performed an enrichment analysis using the KOBAS[41] platform. This analysis identified activated pathways from upregulated genes and suppressed pathways from downregulated genes in MVA-infected cells. (The top 10 pathways are shown in Fig. 5d; see Figures S9 and S10 for the extended list). MVA infection resulted in downregulation of genes within several pathways associated with cell proliferation including the TGFβ pathway, Hippo pathway (which includes WNT-related genes), and the PI3K-Akt pathway, all of which are heavily influenced by viral infection[43–45]. Pathways involving upregulated genes included those associated with carbohydrate metabolism and insulin secretion, both of which are associated with the Warburg effect, an expected outcome of viral infection[46].

## Functional response correlates with architecture and accessibility

The chromatin accessibility profiles of several genes with differential expression displayed altered accessibility as infection progressed. For example, the promoter region of ZNF467 had increased accessibility (infection-biased OCRs) and associated increased gene expression (Fig. 6a). Similarly, the promoter region of *OTOP2* and *USH1G* (nearby genes on sense and antisense strands) with increased accessibility (infection-biased OCRs) had concurrently modest increased gene expression (Fig. 6b). Genes that overlapped with control-biased OCRs (regions with reduced accessibility due to MVA infection) included *IL1R1* (Fig. 5c), *LFNG*, and *TTYH3* genes (Fig. 6d), all of which were downregulated in MVA-infected cells. These genes are associated with cellular immune responses, including the progression of viral infection involving the O-glycan pathway, which was also predicted in the ATAC-seq KOBAS enrichment analysis. Closer inspection of chromatin contacts around these genes suggested chromatin de-regulation due to infection (Fig. 6, Fig. S11); indeed, loss of transcription may have led to inter-domain condensation. For example, in the regions containing the *IL1R1*, *LFNG*, and *TTYH3* genes (Fig. 6c, d), we found increased mixing of topologically associated domains (TADs), which are denoted as triangular features in contact frequency heatmaps. Mixing of TADs resulted in a broader region with higher contact frequency and an associated decrease in accessibility in MVA-infected cells. In a more generalized analysis, we aggregated all TADs per timepoint and identified an overall weakening of *cis* interactions within TADs due to infection, particularly with prolonged infection (Fig. 6e, f, Fig. S12 A). The insulation scores of TAD boundaries were mildly impacted (Fig. 6g) even as infection progressed (Figure S12 b), suggesting that TAD mixing and intra-TAD weaking occurred with minimal change in TAD boundaries. Further, we found that gained/restructured loop domains in MVA-infected cells were larger in size than lost/weakened loops (median size of 400 kb instead of 200 kb in control cells). Furthermore, 64.7% of strong OCRs with increased accessibility and 72% of strong OCRs with reduced accessibility in 24 hpi cells overlapped with re-structured loop domains (empirical $p$-value < 0.001 with 1.21- and 1.41-times increased association than expected by chance respectively). Along the same lines, we find that 67.2% of upregulated genes and 70.3% of downregulated genes overlapped with re-structured loop domains (empirical $p$ value < 0.001 with 1.38- and 1.22-times increased association than expected by chance, respectively). These findings suggest a connection between gene expression changes and loop restructuring induced by MVA infection. Collectively, as infection progressed, differential gene expression was associated with concurrent alterations in 3D chromatin architecture and accessibility.

## Discussion

The architecture of the genome, including its compartmentalization, chromatin contacts, conformation, and accessibility, is as fundamental to understanding the function of a particular genomic locus as its sequence. Indeed, investigation of 3D chromatin structure may be paramount to deciphering disease-states, aberrant cellular behavior, and the impact of environmental perturbations, including pathogen exposures, all of which have functional implications, including dysregulation of gene expression[13,47–50]. Further, pathogen-induced changes in host chromatin features may have far-reaching implications for development of appropriate and efficacious counter measures including vaccine development. Although it is apparent that genome architecture is highly dynamic, the correlation between its structural organization at various scales and functional regulation is still unclear and requires comprehensive empirical data from various

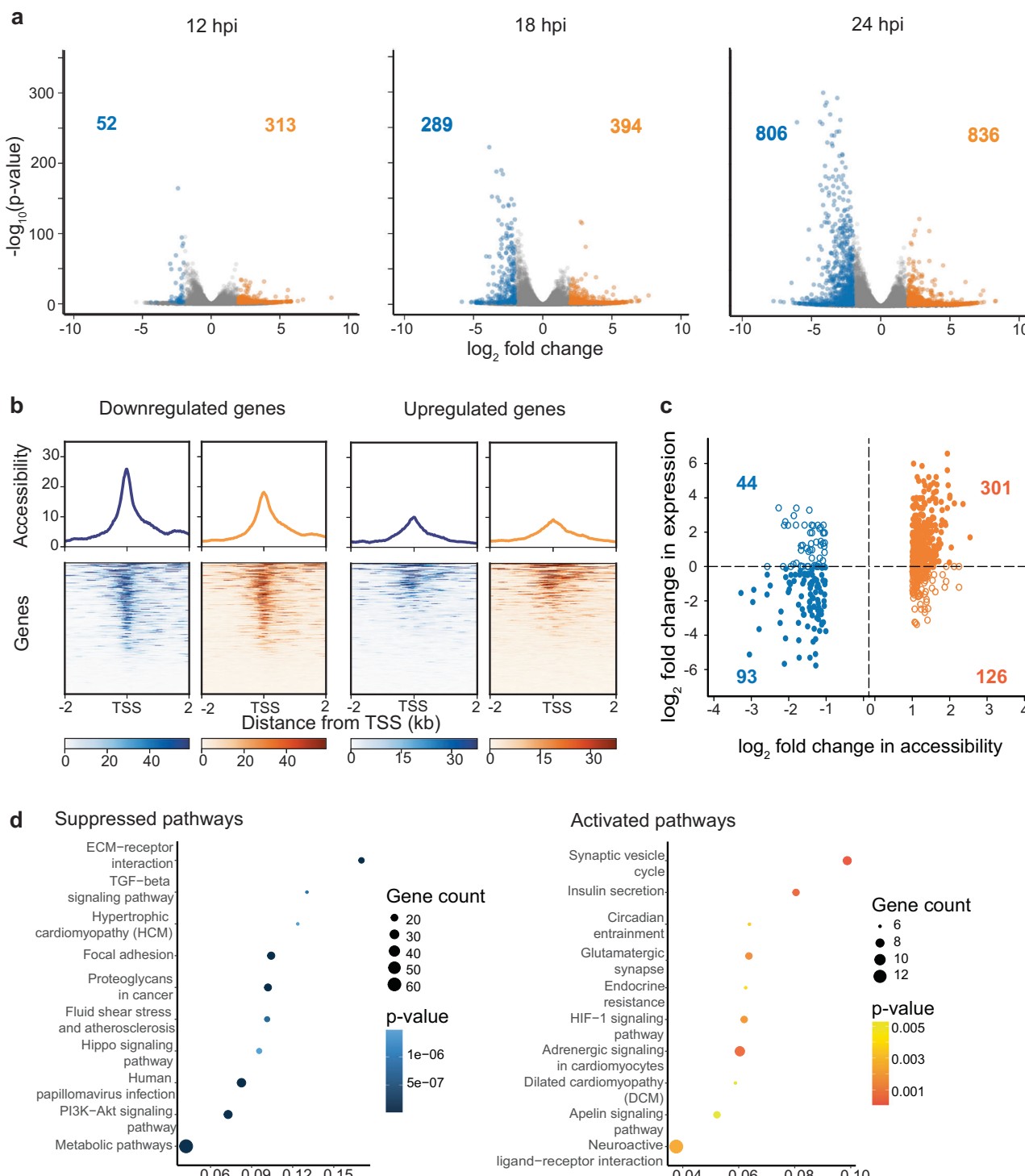

**Fig. 5 | MVA infection induced gene expression changes and moderately correlates with local chromatin accessibility. a** Volcano plots demonstrate differentially expressed genes between mock-infected control and MVA-infected Vero cells at each time point: 12, 18, and 24 hpi. Gray dots represent all genes. Blue dots indicate significantly different (adjusted $p$ value < 0.05) down regulated genes ($\log_2$ fold change < −2), while orange dots indicate significantly upregulated genes ($\log_2$ fold change > 2). **b** Chromatin accessibility around TSS (± 2000 bp) of strongly downregulated (left two panels) and upregulated (right two panels) genes at 24 hpi is plotted. Blue indicates accessibility data from mock-infected control Vero cells and orange indicates MVA-infected cells. The ATAC-seq reads from all biological replicates at 24 hpi are combined. The normalized average read count profile is plotted in the top panel, while the bottom panel shows the normalized read count around TSS where each row represents one consensus gene. **c** $\log_2$ fold change in accessibility at promoter-annotated OCRs (OCRs overlapping with TSS ± 500 bp) vs $\log_2$ fold change in expression in corresponding genes is plotted. OCRs accessible in control cells are shown in the negative x-axis (blue colored) and OCRs accessible in MVA-infected cells are shown in the positive x-axis (orange colored). Positive fold change in gene expression indicates upregulated genes in infected cells and vice versa. **d** Top gene regulatory pathways that are affected by MVA infection predicted by Kegg Orthology analysis of differentially expressed genes. Top 10 suppressed pathways (left panel) and activated pathways (right panel) are shown in the decreasing order of their gene enrichment ratio.

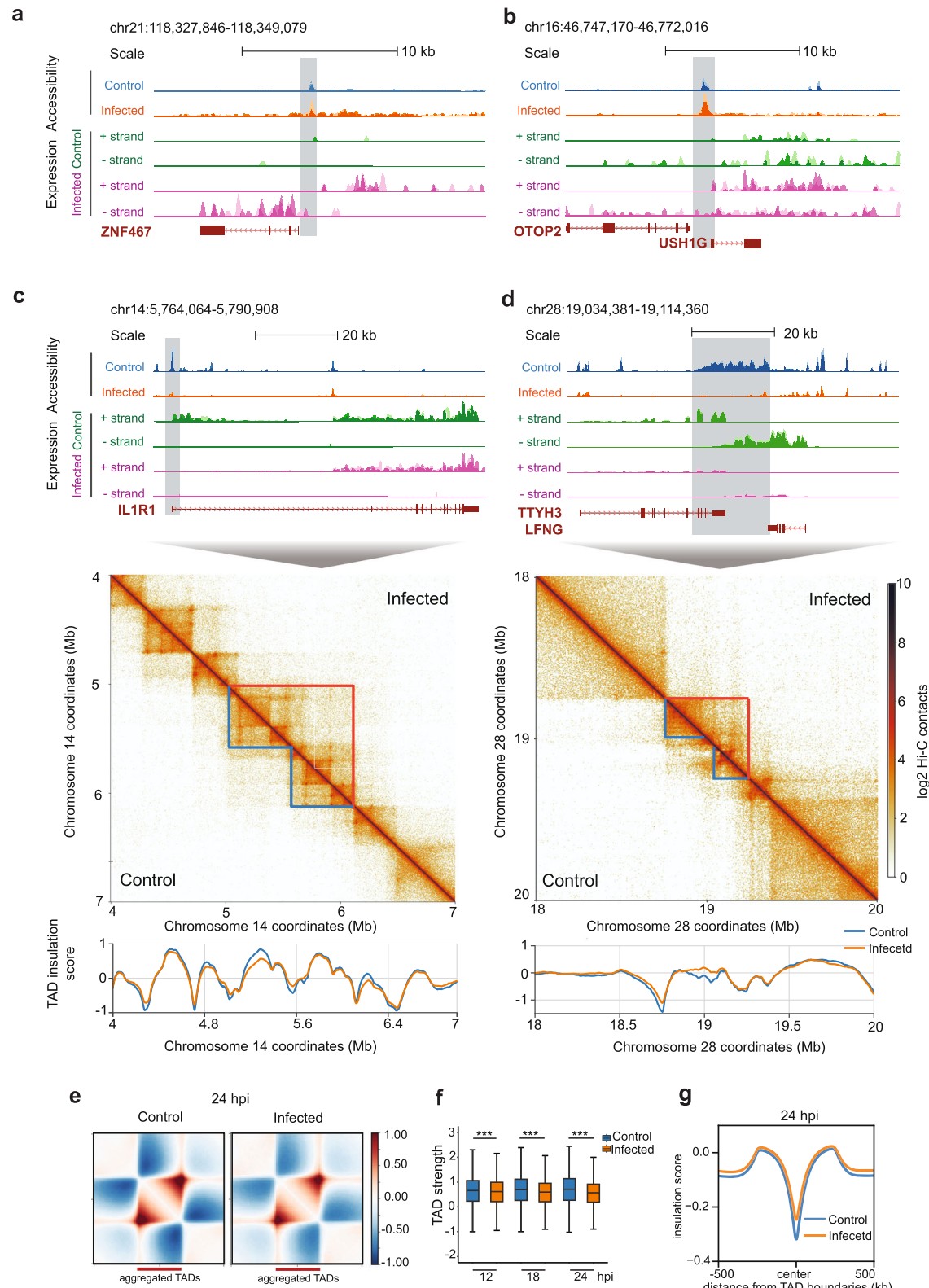

perturbation experiments. To this end, we employed a model system of infection (MVA in Vero) for time course chromatin assessments in a paired manner allowing for direct comparison of global and local architectural features and functional changes.

We found substantial changes in global and local chromatin architecture in infected host genomes. There was an overall increase in mid- to

long-range intra-chromosomal interactions in the host genome and prolonged infection increased changes in contact frequency. We speculate these viral-induced changes in mid- to long-range intra-chromosomal interactions likely represent sub-compartment repositioning. We detected a small fraction of intra-chromosomal interaction changes that localized to fine-scale loop anchors, marking differential loops after infection. Thus,

**Fig. 6 | Functional responses correlate with changes in chromatin architecture and accessibility. a, b** ATAC-seq and RNA-seq signal tracks from 24 hpi at example regions with infection-biased accessibility and gene expression are shown. Replicate signals are overlaid. Blue: accessibility in mock-infected control cells, Orange: accessibility in MVA-infected cells, Green: control gene expression, Purple: infected gene expression. **c, d** ATAC-seq and RNA-seq signal tracks from 24 hpi at example regions with control-biased accessibility and gene expression are shown in the top panel (color scheme the same as **a** and **b**). The corresponding Hi-C contact heat map of a larger region including the corresponding differential accessible window is shown in the middle panel. Control and MVA-infected contacts are shown in the lower and upper diagonal parts of the figure, respectively. Blue and orange markings highlight contact domains with increased mixing in MVA-infected cells. The TAD insulation score for the corresponding region is shown in the bottom panel. Control (blue) and infected (orange) TAD insulation scores are overlaid. **e** Control (left) and infected (right) average log2 observed over expected contact frequency within scaled aggregated TADs at 24 hpi are shown as heatmaps. TAD location is denoted by the maroon bar and additional same sized flanking regions are included on both sides. **f** Comparison of TAD strength between control and infected cells at all three timepoints shows significant weakening of infected TADs. Welch two sample two-tailed $t$ test $p$ value: 5.053e-11, < 2.2e-16, < 2.2e-16 and df 25224, 23776, 26407 respectively at 12, 18, and 24 hpi. The central horizontal line in box plot mark median TAD strength per category. Outliers are omitted from the plot. **g** A profile plot of insulation scores (control: blue, infected: orange) computed from all TAD boundary midpoints ± 500 kb region identified at 24 hpi.

prolonged infection caused moderate fine-scale chromatin remodeling. In fact, gained/strengthened loops due to infection were larger in size compared to lost/weakened loops suggesting that chromatin loop domain restructuring may propagate to larger chromatin architectural features. Indeed, there was an overall weakening of topologically associated domains (TADs) with prolonged infection. This reduced contact frequency within TADs combined with maintenance of TAD boundaries was concurrent with the increased size of extruded loop domains. Similar observations have been reported in response to SARS-CoV-2 infection, potentially due to decreased intra-TAD cohesin binding[13]. Thus, the CTCF/cohesin complex may be involved in the host chromatin restructuring in response to vaccinia virus infection as well. Investigation of this complex and associated histone modifications would provide deeper insight into mechanisms of fine-scale chromatin restructuring in response to infection.

Concurrent with the increased intra-chromosomal interactions, 12% of the Vero genome exchanged compartments with a slight bias from activate (A) to inactive (B) conversion. In many organisms, the extent of compartment exchange is based on cellular phenotype and biological processes; for example, during cellular differentiation in human cells, approximately 36% of the genome strongly exchanges compartments[11], while 60% of the genome is dynamically compartmentalized between various cell types and tissues[51]. In response to the cytoplasmic RNA virus SARS-CoV-2, 30% of the human host genome (carcinomic lung epithelial cells, A549) either exchanges or weakens compartments[13]. In comparison, our findings suggest that MVA infection moderately converts the host Vero genome into a less 'active' mode over time, with greater propensity toward B compartmentalization. Increased contact frequency coupled with relatively few short-range interaction changes suggests that infection induced chromatin condensation due to newly formed larger loops with more contacts, which may reduce accessibility in the nuclear space.

Indeed, at the fine scale, infection reduced accessibility over time across the Vero genome. At any given point, we determined only ~5.1% of the Vero genome is accessible regardless of treatment; less than nine percent of these accessible regions are differentially accessible due to infection (at 24 hours). While this represents a small fraction of the genome, our integrated analysis of chromatin accessibility with differential chromatin loops and differential gene expression suggests that accessibility change at anchor intervals and promoters is correlated with the ability to form loops and modulate gene expression levels, respectively. Gene downregulation was strongly correlated with reduced promoter accessibility more so than upregulation was with increased accessibility. Further, reduced accessibility was observed at anchor points of loops that were lost during infection; yet, infected cells retained the same amount of accessibility (as controls) in loops that were gained. While we did not perform causality studies to conclude whether loss of accessibility precludes loop loss, our results demonstrate that lack of a loop does not cause loss of accessibility, but loss of accessibility is correlated with loop loss. Therefore, we speculate that anchor point accessibility is a predetermining factor for loop formation. The functional relevance of chromatin loop reorganization is predicated on whether loops provide proximal interactions for genes with their enhancer or repressor elements. Further, reorganization may also control localization of genomic elements in the nuclear space thereby causing fine-scale functional changes or vice versa. In agreement with this assertion, we found that differential accessible regions and differentially expressed genes overlapped with restructured loop domains more than expected by chance.

At the functional level, direct examination of gene expression showed downregulation of many genes associated with pathways including host immunity and proliferation in response to viral infection, whereas those associated with cell signaling and carbohydrate metabolism were upregulated; these hallmark expression changes in response to viral infection have been demonstrated by many other studies[52,53]. In addition, our paired data revealed concurrent alteration in chromatin accessibility and contacts at several of those loci, particularly at genes that were strongly downregulated. Several reports provide evidence of viral hijacking of host epigenetic modifications, which have profound influence on gene expression and chromatin contacts; as such, MVA-induced alterations in epigenetic modifications may underscore the changes we observe in gene expression which influence accessibility and chromatin architecture. Characterization of epigenetic changes was not within the scope of this study; however, future studies including comprehensive epigenome profiling would greatly advance our understanding. Further, drug or CRISPR-mediated global or locus-specific perturbation of transcription and/or epigenetic modifications would provide more insights into the directionality of this relationship.

While correlated dynamics were observed among chromatin accessibility, gene expression, and chromatin loops, differentially accessible regions and expressed genes only marginally overlapped with exchanging compartments. We anticipated that if the same factors responsible for large-scale compartment exchange were driving fine-scale accessibility changes as well, then accessibility would follow changes in compartmentalization. On the contrary, our finding indicates that fine-scale changes in nucleosome positioning in response to infection is likely independent of broad-scale chromatin compartmentalization mechanisms. Taken together, increased B compartmentalization, increased intra-chromosomal interactions with decreased inter-chromosomal interactions, and reduced chromatin accessibility suggest that MVA infection converts the host genome into a more condensed state. However, overall structural condensation did not correspond to an overall downregulation of gene expression resulting from MVA infection; rather, changes in gene expression occurred nearly equally in both directions. While stronger directional responses may be elicited by more virulent pathogens, we still observed these fine-scale structural and functional changes with our attenuated, non-integrating virus. As such, we recommend performing host chromatin accessibility profiling to aid in the prediction of cellular responses to vaccines and facilitate optimized vaccine development.

Vaccinia virus belongs to the order Poxviridae, the only cytoplasmic dsDNA viruses, which has not been well studied for impacts on chromatin structure. However, there are several reports on *nuclear* dsDNA viruses, including hepatitis, herpes virus, and adenovirus demonstrating significant effects on host chromosomes. These effects include direct interaction in the active (A) compartment for enhanced viral transcription[54] and significant influence on chromatin compaction[55], accessibility[56], and restructuring for viral replication[57]. While RNA viruses primarily replicate in the cytoplasm, retroviruses spend portions of the viral life cycle in the nucleus and can alter host genome architecture; for example, human T cell leukemia virus type1

(HTLV-1) possesses CTCF binding sites to restructure host genomes to improve viral outcome[58]. Many (cytoplasmic) RNA viruses influence chromatin remodeling via viral proteins: NS1 from influenza A virus inhibits termination of transcription and interferes with cohesin binding at TAD boundaries. Similarly, while vaccinia virus conducts a non-permissive lifecycle in the cytoplasm of host cells[22], vaccinia-derived proteins evade innate immunity and suppress antiviral pathways[59]. Further, a highly virulent strain of vaccinia (not used in this study) promotes hetero-chromatin formation mediated by the viral protein K7 via nuclear translocation[60]. While the mechanism of MVA's influence on chromatin structure remains to be determined, here, we demonstrate that intra-TAD cohesin depletion and gene expression changes may facilitate restructuring of chromatin accessibility and conformation.

Overall, our empirical observations support the hypothesis that genome structure does not determine function in a discrete manner, but rather, promotes and supports function[61]. Thus, Louis Sullivan's famously quipped 'form follows function' not only applies to late 19th century architecture, but also to the impact of genome function on structure further underscoring the complexity of this relationship.

## Methods

### Cell culture and seeding for infection
Cryopreserved Vero cells, derived from the kidney tissue of a normal adult African green monkey, were sourced from the ATCC (Cat # CCL-81). Cells were thawed, passaged, expanded, and seeded for viral infection in complete medium consisting of Dulbecco's Modified Eagle's Medium (DMEM) containing 4 mM L-Glutamine (ATCC, Cat # 30-2002), supplemented with 10% (v/v) Fetal Bovine Serum (FBS, ATCC, Cat # 30-2020) and filter-sterilized before use. Vero cells were maintained at 37 °C with 5% $CO_2$ and 95% relative humidity. Vero cells were cryopreserved at passage #7 from the original source vial using freeze medium consisting of DMEM supplemented with 10% (v/v) FBS and 5% (v/v) dimethylsulfoxide (DMSO, ATCC, Cat # 4-X).

Cryopreserved Vero cells were thawed at passage #8 and propagated for 2 passages prior to seeding for viral infection. At the time of seeding, a cell suspension was made in pre-warmed complete medium and digitally counted using the Scepter 3.0 Handheld Cell Counter and 60 μm Counter Sensors (Millipore, Cat # PHCC360KIT). Cells were seeded for viral infection at passage #11 into replicate tissue culture flasks and multi-well plates using $5 \times 10^6$ cells per T175 flask or $5 \times 10^5$ cells per well in 12-well plates. Seeded flasks and plates were incubated for 24 h at 37 °C, yielding $\sim 1 \times 10^7$ cells per flask and $\sim 1 \times 10^6$ cells per well at the time of infection.

### Viral infection and cell harvesting
Modified vaccinia Ankara virus, strain VACV-MVA, was obtained from the ATCC (Cat # VR-1508). VACV-MVA viral stock was diluted in infection medium consisting of DMEM supplemented with 2% (v/v) FBS. Vero cells were infected at 0 h using 0.4 MOI in a minimal volume of inoculum (3 mL per flask) for 1 h at 37 °C; 17 mL infection medium was added prior to incubation for 12–24 h at 37 °C. In parallel to the flasks, 12-well plates for ICC staining and cell imaging were also infected at 0 hr using 0.4 MOI in 300 μL per well; then 700 μL infection medium was added prior to incubation for 12–24 h at 37 °C.

Cells were harvested in two biological replicates (A and B) per condition (mock-infected and MVA-infected) at each timepoint (12 h, 18 h, and 24 h post-infection). At each timepoint, cells were washed using 1X phosphate-buffered saline (PBS, Gibco, Cat # 10010049), detached for 15 min at 37 °C using 1X TrypLE Express Enzyme (Gibco, Cat # 12604021), and resuspended in pre-warmed complete medium before cell counting with the Scepter 3.0 Handheld Cell Counter and 60 μm Counter Sensors. At each timepoint, mock-infected and MVA-infected cell suspensions were divided and processed for Hi-C, ATAC-seq and RNA-seq techniques (described in section: Library preparation: Hi-C, ATAC-seq, and RNA-seq).

### Immunocytochemistry (ICC) staining and cell imaging
At 12, 18, and 24 h post-infection, a 12-well plate of Vero cells was washed with 1X Hank's Balanced Salt Solution (HBSS) containing calcium and magnesium (Gibco, Cat # 14025092), then cross-linked for 15 min using a 4% (w/v) solution of methanol-free formaldehyde in PBS (Invitrogen, Cat # FB002). Cross-linked cells were washed twice in 1X PBS, then stored in 1X HBSS at 4 °C for 1–2 days. Subsequent ICC staining steps were performed at room temperature. Cells were quenched of residual formaldehyde for 15 min using 0.01% (v/v) Tween-20 in PBS containing 125 mM glycine, then washed twice in 1X PBS. Cells were permeabilized for 15 min using 0.2% TritonX-100 in PBS, then washed twice in 1X PBS. Cells were blocked for 1–2 h using 2% (v/v) Bovine Serum Albumin (BSA, Thermo Scientific, Cat # 37525) + 5% FBS in HBSS, then washed twice in 1X PBS. Five drops of Staining Enhancer solution (Thermo Scientific, Cat # I36933) was added to cells for 30 min, then washed twice in 1× PBS. Cells were stained with a mouse anti-vaccinia virus monoclonal antibody (Santa Cruz, Cat # sc-58210), diluted to 1 μg/mL in 0.02% BSA in HBSS, for 24 h at 4 °C, then washed twice in 0.01% Tween-20 in PBS. Cells were stained with a rabbit anti-mouse IgG-FITC secondary antibody (Santa Cruz, Cat # sc-358916), diluted to 1 μg/mL in 0.02% BSA in HBSS, for 1 h, washed twice in 0.01% Tween-20 in PBS, then washed twice in 1X PBS to remove residual Tween. Cells were stored in SlowFade Diamond Anti-Fade Mountant (Invitrogen, Cat # S36972) at 4 °C. Cell imaging was performed within 2 days of ICC staining using an Invitrogen EVOS FLoid imaging system. Microscopy images were analyzed for total mean fluorescence using ImageJ software[62]. Images were separated into RGB and thresholds were set under "Default" for grayscale coloring for the "green" image. Mean fluorescence intensity for pixels above threshold were measured for each image.

### Library preparation: Hi-C, ATAC-seq, and RNA-seq
For Hi-C, $1 \times 10^6$ cells per aliquot were processed in triplicate on the day of cell harvest. Cells were resuspended in 1X PBS and cross-linked for exactly 10 min using 37% (w/v) formaldehyde containing 10-15% (v/v) methanol (Sigma, Cat # 252549) at a final concentration of 2% (v/v). Formaldehyde was quenched for 5 min using glycine at a final concentration of 0.2 M. Cross-linked cells were incubated on ice for an additional 15 min, then pelleted by centrifugation and washed in 1X PBS. Washed cells were pelleted by centrifugation, and the dry pellets were snap-frozen on dry ice for long-term storage. These cross-linked cells were subsequently processed for Hi-C analysis per the Arima Hi-C Kit manual (Cat # A160134) using the reagents provided in the kit. In summary, cross-linked cells were lysed and enzymatically treated to create proximally-ligated DNA, which was then purified via AMPure XP Reagent (Beckman, Cat # A63880). This proximally-ligated DNA was used to create DNA libraries for sequencing per the Arima Hi-C Library Preparation user guide (Cat # A160141). DNA libraries were then sequenced via the Illumina NextSeq 2000 sequencer platform at the Los Alamos National Laboratory Genome Center.

For ATAC-seq (Assay for Transposase-Accessible Chromatin by sequencing), $2 \times 10^5$ cells were cryopreserved in triplicate 1 mL aliquots using freeze medium on the day of cell harvest. Aliquots were subsequently thawed into complete medium, counted to achieve 80,000 cells per sample, and processed for ATAC-seq analysis per the Active Motif ATAC-seq Kit manual (Cat # 53150) using the reagents provided in the kit. In summary, cells were washed in ice-cold 1X PBS, then lysed in ice-cold "ATAC Lysis Buffer", and centrifuged for 10 min at 4 °C. Tagmented DNA was prepared from the cell lysates and purified via DNA purification beads. DNA libraries were prepared from tagmented DNA amplified via PCR using Q5 Polymerase and i7 and i5 Indexed Primers under the following thermocycling conditions: 72 °C for 5 min, 98 °C for 30 sec, 10 cycles of: 98 °C for 10 sec, 63 °C for 30 sec, 72 °C for 1 min, then 10 °C hold. PCR-amplified libraries were cleaned using magnetic DNA purification beads (provided in kit). Additional bead-based size selection was performed using AmpureXP beads (Beckman, Cat # A63880) following manufacturer's protocol to remove fragments below 150 bp and above 600 bp. DNA libraries were then quantified using a high-sensitivity Qubit dsDNA Quantification Assay kit

(Thermo Scientific, Cat# Q32851), and KAPA Library Quantification Kit (Roche, Cat # KR0405), and sequenced via the Illumina NextSeq 2000 sequencer platform in paired-end mode (PE151).

For RNA-seq, $2 \times 10^5$ cells were cryopreserved in duplicate 1 mL aliquots using freeze medium on the day of cell harvest. Aliquots were subsequently thawed in a 37 °C water-bath into ice-cold complete medium, then pelleted by centrifugation, washed in ice-cold HBSS, and resuspended in 350 µL ice-cold RULT Buffer (Qiagen, Cat # 73934) containing 1% beta-mercaptoethanol (Sigma, Cat #63689). RNA was extracted from these cell lysates using the RNeasy UCP Micro Kit protocol and reagents (Qiagen, Cat #73934). DNA was digested via on-column DNase I treatment for 15 min at room temperature. RNA extracts were eluted in 20 µL RNase-free water. The concentration of total RNA was determined using the Qubit RNA HS Assay Kit (ThermoFisher Scientific, Cat # Q32855) and quality was measured using the Bioanalyzer RNA 6000 Assay (Agilent, Cat # 5067-1511). Ribosomal RNA was depleted from 500 ng total RNA using the Illumina Ribo-Zero Plus rRNA Depletion Kit (Illumina, Cat. # 20037135). RNA was converted to cDNA, and adapters and indexes were added onto the ends of the fragments to generate strand specific Illumina libraries. Libraries were eluted in DNA Elution Buffer (Zymo Research, Cat #D3004-4-10). The concentration and average size of the libraries was verified using the TapeStation DNA 5000 Assay Reagents and Screen Tapes (Agilent, Cat. # 5067-5589 and 5067-5588). Libraries were normalized and pooled based on the TapeStation concentrations. An accurate library quantification of the pool was determined using the Illumina/Universal Kit (KAPA Biosystems, KK4824). The pool was sequenced on an entire lane of a NextSeq 500 HO flow cell to generate paired-end 151 bp reads using the NextSeq 500 High Output Kit v2.5 (300 cycles) (Illumina, Cat # 20024908).

### ATAC-seq analysis

Raw reads from ATAC-seq assays were processed using methods described in ref. [63]. Briefly, reads were trimmed and filtered with fastp[64] to remove Nextera adaptors and reads with repetitive sequences. Reads were also filtered to remove bases with low quality scores ($q < 15$). These processed reads were aligned using bwa[65] to the *Chlorocebus sabaeus* reference genome (GCF_000409795.2_Chlorocebus_sabeus_1.1_genomic.fasta) downloaded from the NCBI genome browser. Prior to alignment, the reference fasta file was modified to include the sequence of the Vaccina virus (Accession No. U94848, downloaded from GenBank). After alignment, duplicate sequenced pairs and mitochondria read pairs were marked and removed from analysis via samblaster[66]. After filtering, sample alignments were analyzed to identify loci displaying significant enrichment of paired-end reads (peaks) using MACS2[67]. Input alignments were further filtered using samtools[68] with the following samtools flags: -F 4 -F 256 -F 512 -F 1024 -F 2048 -q 30 and then passed to MACS2. MAC2 was called in BAMPE mode, with a min-length and max-gap setting of 100 and 150 bp, respectively, in addition to the calling summits option. Replicate correlation analysis was performed with deepTools[69]—plotCorrelation function using MACS2 output bigwig files excluding vaccinia virus contig as input.

For all further analysis, the entire list of MACS2 called peaks except the ones in the vaccinia virus contig were used. The list of open chromatin regions (OCRs), defined as regions that are accessible in at least one of sample studied, were created by taking a union of all MACS2 called peaks followed by merging them using BEDtools[70] merge function. To visualize global changes in accessibility, deepTools –computematrix function with –binsize 10 kb was used to calculate read coverage around all OCR midpoints. Further, plotHeatmap and plotProfile functions were used for visualization. Differential accessibility analysis across all OCRs was performed using the DESeq2[71] matrix design (~time + treatment).

### Hi-C contact map generation and analysis

Hi-C paired fastq.gz files from paired-end Illumina sequencing were analyzed using the Juicer Hi-C pipeline[72]. Samples were aligned to the modified *C. sabaeus* reference genome (described above). Restriction sites associated with the Arima protocol (used to construct Hi-C libraries) were identified

using Juicer. Hi-C files (.hic) were constructed using default settings within Juicer, and for subsequent analysis, the mapping quality threshold of "q = 30" was used. The number of valid Hi-C contacts across samples, after alignment and filtering, ranged from approximately 256 to 534 million (Supplementary Table 1). The spectral statistic[73] was utilized to measure reproducibility between Hi-C replicates (Fig. S2B). The Hi-C replicates demonstrated broad biological reproducibility as demonstrated by this statistic (median scores > 0.75). Replicate Hi-C maps were merged using the "mega" function in Juicer. The number of Hi-C contacts of these merged Hi-C maps ranged from ~687 to 854 million.

### Distance decay and PCA

The calculations of decay frequencies in Hi-C contacts across genomic distances and principal component analysis were conducted in Python using the FAN-C package[74]. Decay profiles were calculated for each chromosome and combined for visualization and analysis. For principal component analysis, a resolution of 50 kb was used to construct genomic bins and the bins with zero Hi-C contacts across all the replicates were removed (-z flag in FAN-C).

### Loop identification

Chromatin loops were identified across the replicate and merged Hi-C maps in Juicer via the HiCCUPs[72] algorithm with default settings at resolutions of 5, 10, and 25 kb. To identify differential loops, Hi-C contact counts were gathered for each replicate across the merged list of loops. These counts were fed into DESeq2[71] to identify the following: 1) differential loops between mock-infected control and MVA-infected libraries and 2) loops with differential Hi-C counts across time. All analyses were performed across the time course (12, 18, and 24 hpi) and separately within a given timepoint.

### A and B compartment assignment and exchange estimation

For each replicate Hi-C map, the initial A and B compartment scores across chromosomes (at a resolution of 100 kb) were estimated using the eigenvector function in Juicer. These results were combined with the replicate ATAC-seq profiles to correctly label active (A) and inactive (B) compartments. Using the convention that the active A compartment should have more ATAC-seq peaks, the number of ATAC-seq peaks in every 100 kb bin were correlated with the associated A and B scores. For each chromosome, compartment scores were reoriented (i.e., multiplying scores per chromosome by negative one) if the 95th percentile of counts of ATAC-seq peaks were labeled as compartment B in a majority (four out of the six) of ATAC-seq libraries. To identify compartment exchange due to MVA infection, we selected 100 kb bins assigned to the same compartments in both Hi-C biological replicates. Then, the replicate average compartment score was determined for each bin considering only the bins in agreement between replicates. The changes from positive (in control) to negative (in infected) bin score value were assigned as A to B exchange; similarly, changes from negative (in control) to positive (in infected) bin score value were assigned as B to A exchange. Bins that remained positive or negative were assigned as "no exchange" A or B compartments, respectively.

### Inter-chromosome interaction analysis

An inter-chromosomal interaction score was calculated for each pair of chromosomes within each Hi-C experimental condition. As described in detail in ref. [75], this score is a ratio calculated by dividing the number of Hi-C contacts between a pair of chromosomes by the expected frequency of inter-chromosomal Hi-C contacts involving those chromosomes. After calculating this ratio for each Hi-C map, the difference (in $\log_2$ scale) in inter-chromosomal score was calculated across the infection time course (12, 18, and 24 hpi) between each pair of mock-infected control and MVA-infected maps.

### Differential interaction analysis

multiHiCcompare[36] was used to perform joint normalization (using fastlo method) of the Hi-C matrix across comparative datasets and to identify

differential contacts (using hic_exactTest). For input, the Hi-C contact matrix converted to a four-column bed file (chromosome, region1, region2, IF) and 10 kb contiguous windows across the genome were used.

## TAD strength and insulation score analysis

Topologically associated domains (TADs) were called from in replicate combined Hi-C maps using the arrowhead function from the Juicer toolbox[72]. The arrowhead function was evoked on Knight-Ruiz balanced[76] Hi-C maps at 10 kb resolution by adding the "-r 10000" and "-k KR" parameters (respectively). Additionally, the -m parameter was set to 5000 (-m 5000). To perform TAD strength analysis, TADs identified from control and infected samples per time point were combined and duplicates were removed. Aggregate $\log_2$ of observed over control contact frequency (at 10 kb resolution) within these merged, scaled TADs ± same-sized region was plotted using the fanc aggregate function from the FAN-C package[74]. Further, –tad-strength command with –tads flag was used to quantify strength of each TAD in replicate merged control and infected samples separately. TAD insulation scores were calculated using the fanc insulation function from the FAN-C package in 500 kb sliding windows for replicate merged control and infected samples separately using .hic files at 5 kb resolution. TAD boundaries were identified from this insulation score table using the fanc boundaries function which identifies all intervals with insulation score minima as a TAD boundary. These TAD boundary coordinates for control and infected samples at each timepoint were then merged and the insulation score profiles at and around (±500 kb) the midpoints of TAD boundaries were plotted using the deeptools[69] plotProfile function.

## Pathway enrichment prediction based on accessibility

Lists of genes whose promoter (TSS ± 500 bp) overlapped with a strong differential peak (fold change >1 and base mean >100) were prepared. This type of conservative list, predicting altered gene transcription due to infection, was generated for both MVA infection accessible and control accessible genes for each time point. KOBAS[41] software was used to identify the statistically significant KEGG pathways enriched in each list using the entire list of genes in *Chlorocebus sabaeus* as background.

## RNA-seq analysis

The *Chlorocebus sabaeus* reference genome was retrieved from NCBI (GCF_000409795.2) and appended with the vaccinia virus (VACV) genome, retrieved from GenBank (U94848.1). Genome annotations for *C. sabaeus* were downloaded from NCBI and the GenBank flat file annotations for VACV were retrieved using the bio[77] package command (bio fetch U94848.1 --format gff) and converted to gtf format using the AGAT[78] package function (agat_convert_sp_gff2gtf.pl). The *C. sabaeus* annotation was appended with the VACV annotation for subsequent analysis. Raw read quality control was performed using FastQC[79] and SeqMonk[80] to verify library quality and degree of PCR duplication. All metrics were normal. A splice junction aware reference index was generated for the combined genome and annotation files using STAR[81] with the arguments (--runMode genomeGenerate --sjdbOverhang 149). Pre-alignment read trimming was not performed in favor of soft-clipping/quality filtering by the STAR aligner. Reads were aligned to the combined genome using STAR. Duplicate reads were marked but retained in downstream analysis per DESeq2 usage guidelines. Sense strand reads aligning to gene features were counted using htseq-count[82] with the arguments (--stranded=reverse). Antisense strand reads aligning to gene features were counted using htseq-count with the arguments (--stranded=yes). Differential expression analysis was performed using the DESeq2[71] matrix design (~time + treatment). For visualization, replicate alignment files were merged using samtools. bigWigs were then generated using the bamCoverage function from the deepTools[69] package with the arguments (--effectiveGenomeSize 2836634000 --normalizeUsing BPM –filterRNAstrand {forward/reverse/NULL} --binSize 1).

## Gene enrichment analysis

Up and down-regulated genes, with adjusted *p* value < 0.05 and $\log_2$ fold change greater than 2 or less than 2 respectively were used. This type of conservative list, representing altered gene transcription due to infection, was generated for both MVA infection and control datasets for each time point. KOBAS[41] software was used to identify the statistically significant KEGG pathways enriched in each list using the entire list of genes in *Chlorocebus sabaeus* as background.

## Accessibility, gene expression, chromatin loop integration analysis

Accessibility across differential loop anchor intervals and around TSS of differentially expressed genes was calculated using deepTools[69] –computematrix function with –binsize 10 kb, skipping bins with zero read coverage. For estimating average treatment by time, accessibility across all control-biased ($p_{adj} < 0.05$ and $\log_2$foldchange < 0) and infection-biased ($p_{adj} < 0.05$ and $\log_2$foldchange > 0) loop anchors --scale-regions mode was used. Conversely, --referencepoint mode was used to estimate accessibility around TSS of upregulated genes ($p_{adj} < 0.05$ and $\log_2$foldchange > 2) and downregulated genes ($p_{adj} < 0.05$ and $\log_2$foldchange < 2) at 24 hpi. plotHeatmap and plotProfile functions were used for visualization.

To estimate the fraction of differential OCRs and genes that overlap with gained/restructured loop domains due to infection, we considered all infection-biased loops ($p_{adj} < 0.05$ and $\log_2$foldchange > 0, $n = 5973$) and assigned the region between the left boundary of the left anchor interval and the right boundary of the right anchor interval as a gained/restructured loop domain. Further, the BEDtools[70] intersect function was used to estimate the number of decreased accessibility OCRs ($p_{adj} < 0.05$, baseMean >100, $\log_2$foldchange < −1, $n = 5521$) and increased accessibility OCRs ($p_{adj} < 0.05$, baseMean > 100, $\log_2$foldchange > 1, $n = 984$) overlapping with gained/restructured loop domains. Similarly, the overlaps of upregulated genes ($p_{adj} < 0.05$, $\log_2$foldchange > 2, $n = 915$) and downregulated genes ($p_{adj} < 0.05$, $\log_2$foldchange < −2, n = 787) were also calculated. To obtain a null distribution, differential OCRs and genes were shuffled 10,000 times across the entire *Chlorocebus sabaeus* genome using BEDtools shuffle function and the overlap with gained/restructured loop domain with each iteration was calculated. "empPvals" function from R package "qvalue" was used to estimate the associated empirical *p* value.

## Statistics and reproducibility

Methods and tools used for statistical analysis performed in this study are provided in the Methods section or in figure legends. For each experimental data presented, such as ATAC-seq, RNA-seq, Hi-C, and fluorescence imaging, we used two independent biological replicates at each time point to test the statistical significance of the differential signal. DESeq2 was used for identifying statistically significant differential accessibility, differential gene expression, and differential chromatin loops. Statistically significant differential chromatin contacts (differential interaction analysis) were identified using multiHiCcompare. Significance of differential TAD strength was tested by applying Welch two sample two-tailed t-test on replicate merged TAD strength score identified per condition. Significant association of differential OCRs and differential genes with restructured chromatin loop domains were tested by computing empirical *p*-value on number of observed overlap vs number of expected overlap by 10,000 iterations of random feature shuffling across the genome. The "empPvals" function from the R package "qvalue" was used to estimate the associated empirical *p*-value. Differences were considered statistically significant at *p*-values of 0.001, 0.01, and 0.05 and described in the figure legends and results sections.

## Reporting summary

Further information on research design is available in the Nature Portfolio Reporting Summary linked to this article.

## Data availability

All sequencing raw data generated during this study from ATAC-seq, RNA-seq and Hi-C experiments are available in the SRA repository, under the BioProject accession number PRJNA1037174. Processed data files supporting the conclusions of this article are available in GEO repository with accession number: GSE248052. All imaging data is uploaded in figshare with https://doi.org/10.6084/m9.figshare.25582833.

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

## Acknowledgements

Research presented in this article was supported by the Laboratory Directed Research and Development program of Los Alamos National Laboratory under project numbers 20210134ER (CRS and KYS) and 20210082DR (KYS and SS), and by the U.S. Department of Energy, Office of Science, through the Biological and Environmental Research (BER) and the Advanced Scientific Computing Research (ASCR) programs under contract number 89233218CNA000001 to Los Alamos National Laboratory (Triad National Security, LLC) awarded to SS and CRS.

## Author contributions

C.R.S. and V.V. designed the viral experiments and chromatin architecture, accessibility, and expression assays and analyses. S.A. performed the VACV viral infections and microscopy. C.R.S. performed fluorescence image analysis. S.A. and V.V. performed cell harvest and sample collection. S.A. performed the RNA extractions and the LANL Genome Center prepared the RNA-seq libraries. V.V. performed sample processing and preparation of ATAC-seq and Hi-C libraries. All sequencing was conducted at the LANL Genome Center. C.R. aligned and processed the Hi-C sequencing data. C.R. and V.V. analyzed the Hi-C data. C.R. and V.V. developed the pipeline for ATAC-seq analysis. V.V. analyzed the ATAC-seq data, including visualization and interpretation. E.S. processed the RNA-seq data. V.V. interpreted and integrated all datasets (Hi-C, ATAC-seq, and RNA-seq). V.V. and C.R. generated the figures and tables. V.V. and C.R.S. performed analyses and biological interpretation of all datasets and wrote the manuscript. Funding was provided by CRS, SS, and KYS.

## Competing interests

The authors declare no competing interests.
