## [Peer Review File · Communications Biology]

Reviewers' comments:

Reviewer #1 (Remarks to the Author):

In this work, the authors conducted comprehensive multi-omic analyses of host epigenome after acute smallpox virus infection, with a collection of Hi-C, ATAC-Seq, and RNA-Seq datasets. The authors reported significant global and local structural changes caused by viral infection, such as increased long-range chromatin interactions and B-compartmentalization, and decreased inter-chromosomal interactions. They also performed correlational and functional analyses between the changes in chromatin structures, accessibilities, and the changes in gene expression. They concluded that fine-scale chromatin remodelings are more closely associated with gene expression alterations. Overall, this paper was well-written, and experiments and analyses were well-executed. I have only minor comments here that might help to improve this paper:

- 1). For Figure 1B, it is better to have statistical quantifications of how many GFP-positive cells indicate the rate of infection at different time points.
- 2). For Figure 6B, it would be nicer to have shown the Hi-C changes of ZNF467 and USH1G genes as well.
- 3). For discussions, the authors should discuss what are the possible mechanisms of how viral infection causes structural changes, whether it's directly through viral proteins or non-direct cellular responses.
- 4). The authors stated "chromatin accessibility and local structure profiling provide impactful predictions for host responses and optimization of vaccine design", they should elaborate more on this part, about how epigenome profiling can provide predictions and optimizations of the vaccine.

Reviewer #2 (Remarks to the Author):

In their manuscript, Venu and colleagues explore the impact of MVA infection on the architecture and accessibility of the host genome. Employing multi-omics methodologies, they reveal how prolonged infection induces alterations in chromatin structure, which correlate with changes in gene expression.

Overall, the data support the authors' conclusions, and the results are meticulously presented, including appropriate controls. However, the findings tend to be descriptive in nature, lacking detailed mechanistic insights into the observed changes. For instance, considering the known influence of other viruses on cellular factors involved in 3D chromatin regulation, such as CTCF and the Cohesin complex, integrating studies on their binding dynamics during prolonged MVA infection would enhance the significance of this work.

Additionally, the study's significance is lessened by certain limitations. The analysis is restricted to a limited number of time points, raising questions about the persistence of the observed changes over longer durations. Moreover, the potential influence of cell cycle alterations on the observed chromatin changes remains unclear. Furthermore, it would be pertinent to consider whether these chromatin alterations correlate with the cytopathic effects typically associated with MVA infection in Vero cells.

Addressing these limitations, or at least acknowledging them in the discussion section of the manuscript, would enrich the interpretation of the reported results.

Dr. Christina Steadman
Deputy Group Leader & PI
Los Alamos National Laboratory
P.O. Box 1663, MS J495
Los Alamos, NM 87545
505-469-4322

April 10, 2024

Subject: Rebuttal Letter

Dear Reviewers,

We would like to extend our sincerest appreciation for your ample investment in providing a constructive, detailed response to our manuscript. We have sought to improve the manuscript per your suggestions and provide detailed responses (in black) to each suggestion (in blue) below. Where appropriate, we have called out changes in the manuscript (or provided them in this document).

Reviewer #1

(Remarks to the Author): In this work, the authors conducted comprehensive multi-omic analyses of host epigenome after acute smallpox virus infection, with a collection of Hi-C, ATAC-Seq, and RNA-Seq datasets. The authors reported significant global and local structural changes caused by viral infection, such as increased long-range chromatin interactions and B-compartmentalization, and decreased inter-chromosomal interactions. They also performed correlational and functional analyses between the changes in chromatin structures, accessibilities, and the changes in gene expression. They concluded that fine-scale chromatin remodeling are more closely associated with gene expression alterations.

Comments:

Overall, this paper was well-written, and experiments and analyses were well-executed. We truly appreciate the positive comments about our study as we seek to perform rigorous repeatable experiments for the community at large.

I have only minor comments here that might help to improve this paper:

1) For Figure 1B, it is better to have statistical quantifications of how many GFP-positive cells indicate the rate of infection at different time points.

We have now quantified the overall GFP mean fluorescence intensity using ImageJ analysis for the MOI images and the time course (Figure 1B). This analysis is provided in Supplemental Figure 1. Please note that the quantification shows changes in fluorescence intensity as MOI increases and compared with an unstained control. All microscopy images are now uploaded (not made publicly available yet) in figshare with DOI: 10.6084/m9.figshare.25582833.

2) For Figure 6B, it would be nicer to have shown the Hi-C changes of ZNF467 and USH1G genes as well.

Thank you for this suggestion! We looked at these genes in greater detail in all our data sets. While TAD mixing is observed around these genes, the changes are not as evident as regions for ILR1 and TTYH3 (shown in Figure 6D and 6E). Therefore, we have included Hi-C images for ZNF467 and USH1G in Supplemental Figure 11.

3) For discussions, the authors should discuss what are the possible mechanisms of how viral infection causes structural changes, whether it's directly through viral proteins or non-direct cellular responses.

We have added the following sentences in the discussion (lines 567-584 (Final Clean version) and lines 726-746 (Track Changes version)) to discuss what others have found regarding the mechanism of action for vaccinia virus and other types of viruses.

“Vaccinia virus belongs to the order Poxviridae, the only cytoplasmic dsDNA viruses, which has not been well studied for impacts on chromatin structure. However, there are several reports on nuclear dsDNA viruses, including hepatitis, herpes virus, and adenovirus demonstrating significant effects on host chromosomes. These effects include direct interaction in the active (A) compartment for enhanced viral transcription¹ and significant influence on chromatin compaction², accessibility³, and restructuring for viral replication⁴. While RNA viruses primarily replicate in the cytoplasm, retroviruses spend portions of the viral life cycle in the nucleus and can alter host genome architecture; for example, human T cell leukemia virus type1 (HTLV-1) possesses CTCF binding sites to restructure host genomes to improve viral outcome⁵. Many (cytoplasmic) RNA viruses influence chromatin remodeling via viral proteins: NS1 from influenza A virus inhibits termination of transcription and interferes with cohesin binding at TAD boundaries⁶. Similarly, while vaccinia virus conducts a non-permissive lifecycle in the cytoplasm of host cells⁷, vaccinia-derived proteins evade innate immunity and suppress antiviral pathways⁸. Further, a highly virulent strain of vaccinia (not used in this study) promotes heterochromatin formation mediated by the viral protein K7 via nuclear translocation⁹. While the mechanism of MVA's influence on chromatin structure remains to be determined, here, we demonstrate that intra-TAD cohesin depletion and gene expression changes may facilitate restructuring of chromatin accessibility and conformation.”

4). The authors stated "chromatin accessibility and local structure profiling provide impactful predictions for host responses and optimization of vaccine design", they should elaborate more on this part, about how epigenome profiling can provide predictions and optimizations of the vaccine.

Our results show that chromatin accessibility changes are closely related to both functional (transcriptional) and structural (loop organization) changes following the viral infection model studied here. Given the ease of characterizing accessibility via ATAC-seq, one could envision employing it to find the differential effects of certain treatments in addition to impacts of pathogens. This is a less explored area with great potential in vaccine development. However, given that we did not explicitly assess the MVA vector, we simply postulate that as sequencing based screening approaches are being widely

used, chromatin profiling may be a useful tool for vaccine development. This is included in the Discussion (lines 563-566 (Final Clean version); 721-725 (Track Changes version)):

“While, stronger directional responses may be elicited by more virulent pathogens, we still observed these fine-scale structural and functional changes with our attenuated, non-integrating virus. As such, we recommend performing host chromatin accessibility profiling to aid in the prediction of cellular responses to vaccines and facilitate optimized vaccine development.”

Reviewer #2

(Remarks to the Author): In their manuscript, Venu and colleagues explore the impact of MVA infection on the architecture and accessibility of the host genome. Employing multi-omics methodologies, they reveal how prolonged infection induces alterations in chromatin structure, which correlate with changes in gene expression.

Overall, the data support the authors' conclusions, and the results are meticulously presented, including appropriate controls.

Thank you for noticing our effort to present results in this manner!

Comments:

1) However, the findings tend to be descriptive in nature, lacking detailed mechanistic insights into the observed changes. For instance, considering the known influence of other viruses on cellular factors involved in 3D chromatin regulation, such as CTCF and the Cohesin complex, integrating studies on their binding dynamics during prolonged MVA infection would enhance the significance of this work.

We agree and acknowledge that detailed mechanistic investigation of infection induced chromatin changes were out of the scope of this study. We would initially like to focus this manuscript on the detailed investigation of concurrent dynamics of chromatin structural features and functional output at three timepoints following viral infection. We find that as infection progresses (up to 24 hours post infection) the magnitude of chromatin changes increases without severely affecting cell viability. These results allow us to design future experiments to gain mechanistic insights focusing at 24 hpi (for example, knockout of viral VACV proteins to determine their role in chromatin dynamics).

To our knowledge binding dynamics of CTCF/cohesin or other architectural proteins have not been studied to date in this (Vero-MVA) model system. However, several features associated with mechanistic basis of 3D chromatin changes are identified with other viral infections. This includes viral protein mediated transcriptional elongation past TES⁶ and weakening of intra-TAD contacts due to intra-TAD cohesin depletion¹⁰. While the RNA-seq experiment in our study was not particularly designed to capture real time transcription, we do not see any evidence of read through transcript past TES (result not shown in manuscript) minimizing the chance of this possibility. However, we did find differences when we further investigated the changes in chromatin organization by comparing control and infected samples at the TAD level. We performed the following two analyses and results are added into the results section of the revised manuscript,

including 1) comparison of the intra-TAD contact frequency (TAD strength) genome-wide, and 2) the strength of insulation at TAD boundaries. These analyses revealed that there is an overall TAD weakening (reduction in intra-TAD contacts) with largely undisturbed TAD boundaries. Results from these analyses together with our earlier finding of increase in loop-domain size post-infection indicates a possibility of intra-TAD cohesin depletion resulting in increased loop sizes with weakened TAD structure, similar to the observation after SARS-CoV2 infection.

Results of these new analysis are added as Figures 6 E, F, G, Figures S 12 A, and B.

New language and analysis in the results section is as follows (line 432 Final Clean version or line 492 in Track Changes version):

“For example, in the regions containing the IL1R1, LFNG, and TTYH3 genes (Figure 6B), we found increased mixing of topologically associated domains (TADs), which are denoted as triangular features in contact frequency heatmaps. Mixing of TADs resulted in a broader region with higher contact frequency and an associated decrease in accessibility in MVA-infected cells. In a more generalized analysis, we aggregated all TADs per timepoint and identified an overall weakening of cis interactions within TADs due to infection, particularly with prolonged infection (Figure 6E, 6F, Figure S12 A). The insulation scores of TAD boundaries were mildly impacted (Figure 6G) even as infection progressed (Figure S12 B), suggesting that TAD mixing and intra-TAD weakening occurred with minimal change in TAD boundaries.”

New language in the discussion was also included (line 496 Final Clean version or line 566 in Track Changes version):

“Indeed, there was an overall weakening of topologically associated domains (TADs) with prolonged infection. This reduced contact frequency within TADs combined with maintenance of TAD boundaries was concurrent with the increased size of extruded loop domains. Similar observations have been reported in response to SARS-CoV2 infection, potentially due to decreased intra-TAD cohesin binding¹⁰. Thus, the CTCF/cohesin complex may be involved in the host chromatin restructuring in response to MVA infection as well. Investigation of this complex and associated histone modifications would provide deeper insight into mechanisms of fine-scale chromatin restructuring in response to infection.”

2) Additionally, the study's significance is lessened by certain limitations. The analysis is restricted to a limited number of time points, raising questions about the persistence of the observed changes over longer durations. Moreover, the potential influence of cell cycle alterations on the observed chromatin changes remains unclear. Furthermore, it would be pertinent to consider whether these chromatin alterations correlate with the cytopathic effects typically associated with MVA infection in Vero cells.

Addressing these limitations, or at least acknowledging them in the discussion section of the manuscript, would enrich the interpretation of the reported results.

Yes, we agree that the study has these limitations. Three time points with time-matched controls in biological duplicates for each condition for structural (Hi-C, ATAC-seq) and

functional (RNA-seq) data produced a considerable amount of data that was sufficient to at least to initially characterize how cytoplasmic dsDNA viruses impact the host genome architecture. Our focus was to identify early chromatin changes that occur before cell viability is considerably impacted and cytopathic effects were pronounced. Therefore, for the scope of this study, we choose a strategy to obtain genome-wide high-resolution structure and function data from three limited timepoints.

We agree that the cell cycle is a consideration; to address this, we have time-matched controls per time point that are used for all differential calculations (given that they are in the same part of the cell cycle as their infected counterparts). We acknowledge that for the scope of this study, we have not characterized cytopathic effects of MVA infection on Vero cells. However, based on gene expression patterns we have described the potential pathways activated and suppressed due to infection (Figure 5D). We further show that the majority of the gene expression changes are associated with changes in chromatin accessibility and loop domain reorganization indicating that chromatin alterations correlate with cytopathic effects associated with MVA infection. [We have explained this in lines 537-543 of the Final Clean version of the manuscript.] Further, many studies have assessed the impact of Western Reserve VACV which is more virulent than Ankara VACV. Therefore, we were able to choose a low rate of infectivity (MOI 0.4) such that we would avoid substantial cytopathic effects to observe the chromatin changes that are likely upstream (in time) of the longer-term cytopathic effects.

We have added data to demonstrate that MVA-VACV does somewhat impact viability (Figure S1). We have added the following language in the manuscript as well in line 124: *“The use of time-matched cultures control for inherent changes in chromatin structure or function related to time spent in culture that are unrelated to infection progression.”*

Once again, we would like to express our sincere gratitude for the constructive feedback and suggestions. We believe that these revisions have considerably improved our manuscript, and we hope you find the revisions satisfactory as well. We are looking forward to hearing from you!

Sincerely,

Christina Steadman

References

- 1 Moreau, P. *et al.* Tridimensional infiltration of DNA viruses into the host genome shows preferential contact with active chromatin. *Nat Commun* **9**, 4268, doi:10.1038/s41467-018-06739-4 (2018).

- 2 Rosemarie, Q., Kirschstein, E. & Sugden, B. How epstein-barr virus induces the reorganization of cellular chromatin. *Mbio* **14**, doi:10.1128/mbio.02686-22 (2023).
- 3 SoRelle, E. D. *et al.* Epstein-barr virus evades restrictive host chromatin closure by subverting b cell activation and germinal center regulatory loci. *Cell Reports* **42**, doi:ARTN 11295810.1016/j.celrep.2023.112958 (2023).
- 4 Wang, C. *et al.* A DNA tumor virus globally reprograms host 3d genome architecture to achieve immortal growth. *Nature Communications* **14**, doi:ARTN 159810.1038/s41467-023-37347-6 (2023).
- 5 Satou, Y. *et al.* The retrovirus htlv-1 inserts an ectopic ctf-binding site into the human genome. *P Natl Acad Sci USA* **113**, 3054-3059, doi:10.1073/pnas.1423199113 (2016).
- 6 Heinz, S. *et al.* Transcription elongation can affect genome 3d structure. *Cell* **174**, 1522-+, doi:10.1016/j.cell.2018.07.047 (2018).
- 7 Hruby, D. E., Guarino, L. A. & Kates, J. R. Vaccinia virus replication. I. Requirement for the host-cell nucleus. *J Virol* **29**, 705-715, doi:10.1128/JVI.29.2.705-715.1979 (1979).
- 8 Smith, G. L., Talbot-Cooper, C. & Lu, Y. X. How does vaccinia virus interfere with interferon? *Advances in Virus Research, Vol 100* **100**, 355-378, doi:10.1016/bs.aivir.2018.01.003 (2018).
- 9 Teferi, W. M. *et al.* The vaccinia virus k7 protein promotes histone methylation associated with heterochromatin formation. *PLoS One* **12**, e0173056, doi:10.1371/journal.pone.0173056 (2017).
- 10 Wang, R. *et al.* Sars-cov-2 restructures host chromatin architecture. *Nat Microbiol* **8**, 679-694, doi:10.1038/s41564-023-01344-8 (2023).

REVIEWERS' COMMENTS:

Reviewer #1 (Remarks to the Author):

The revised manuscript has been improved and addressed my previous comments. I recommend publication.

Reviewer #2 (Remarks to the Author):

In the revised manuscript, the authors addressed the previous concerns.